# Profile-based Estimated Inversion Strength

Zhenquan Wang[1], Jian Yuan[1, *], Robert Wood[2, *], Yifan Chen[1] and Tiancheng Tong[3]

[1] School of Atmospheric Sciences, Nanjing University, Nanjing, China

[2] Department of Atmospheric Science, University of Washington, Seattle, USA

[3] Tianwen School, Yichang, China

*Correspondence to:* Jian Yuan (jiany@nju.edu.cn) and Robert Wood (robwood2@uw.edu)

**Abstract.** To better measure the planetary boundary layer inversion strength (IS), a novel profile-based method of estimated inversion strength (EIS$_p$) is developed using the ERA5 daily reanalysis data. The EIS$_p$ is designed to estimate the IS based on the thinnest possible reanalysis layer above the lifting condensation level encompassing the inversion layer. At a ground-based
site in south America, the EIS$_p$ better correlates with the radiosonde-detected IS (R=0.74) than the lower-tropospheric stability (LTS, R=0.53) and the estimated inversion strength (EIS, R=0.45). And the daily variance in low-cloud cover (LCC) explained by the EIS$_p$ is twice that explained by the LTS and EIS. Higher correlations between the EIS$_p$ and the radiosonde-detected IS are also found at other radiosonde stations of subtropics and midlatitude.

Analysis of LCC observed by geostationary satellites and the Moderate Resolution Imaging Spectroradiometer shows
that the EIS$_p$ explains 78% of the annual mean LCC spatial variance over global oceans and land, larger than that explained by the LTS/EIS (48%/13%). Over tropical and subtropical low-cloud prevailing eastern oceans, the LCC range is more resolved by the EIS$_p$ (48%) than the LTS/EIS (37%/36%). And the EIS$_p$ explains a larger fraction (32%) in the daily LCC variance, as compared to that explained by the LTS/EIS (14%/16%). The seasonal LCC variance explained by the EIS$_p$ is 89%, larger than that explained by the LTS/EIS (80%/70%). The LCC-EIS$_p$ relationship is more uniform across various time scales than the
LCC-LTS/EIS relationship. It is suggested that the EIS$_p$ is a better cloud controlling factor for LCC and likely a useful external environmental constraint for process-level studies in which there is a need to control for large-scale meteorology in order to isolate the cloud responses to aerosols on short timescales.

## 1. Introduction

The inversion strength (IS) of the planetary boundary layer (PBL) is an important factor affecting PBL moisture trapping
and low cloud formation. Strong IS inhibits the dry air above the inversion from being incorporated into the PBL and traps moisture below the inversion to favor greater cloud cover (Wood and Bretherton, 2006; Mauger and Norris, 2010). In contrast, weak IS promotes the drying effect of entrained air from the free troposphere and reduces the PBL moisture to decrease cloud cover (Bretherton and Wyant, 1997; Myers and Norris, 2013). Currently two approximate measures of the IS based on reanalysis data are widely used as meteorological constraints on low cloud cover (LCC): the lower-tropospheric stability (LTS,
Klein and Hartmann (1993)) and the estimated inversion strength (EIS, Wood and Bretherton (2006)). They are both defined as a two-level potential temperature ($\theta$) difference between the 700hPa level and the surface but for the EIS the moist adiabatic $\theta$ increase above the lifting condensation level (LCL) is removed in addition. The EIS can be combined with the moisture difference between the 700hPa and surface to form a new stability index, the estimated cloud-top entrainment index (ECTEI). The ECTEI and the EIS have similar correlations with LCC on the seasonal time scales (Kawai et al., 2017).

The LTS and EIS are the best known and most widely-used cloud controlling factors to explain LCC variations. Enhanced LTS can moisten PBLs and has been shown to precede LCC changes by about 24-36 hours (Mauger and Norris, 2010; Klein, 1997). Similarly, Myers and Norris (2013) found that the EIS is the main cause of LCC variations and enhanced subsidence actually decreases LCC for the same value of the EIS. This LCC-LTS/EIS relationship is vital for not only separating observational aerosol effects on clouds from meteorological influences (L'ecuyer et al., 2009; Rosenfeld et al., 2019; Murray-

Watson and Gryspeerdt, 2022; Coopman et al., 2016) but also estimating low cloud climate feedbacks (Klein et al., 2017; Sherwood et al., 2020). In terms of aerosol-cloud interactions, the LTS and EIS can be used to constrain meteorological influences and thus largely reduce the confounding influence of meteorology to separate aerosol effects on low clouds (Mauger and Norris, 2007; Coopman et al., 2016), since LCC variations are most explained by the LTS/EIS among all of LCC-controlling meteorological factors (Stevens and Brenguier, 2009). Without strong cloud-controlling factors, the confounding influence of meteorology is poorly constrained and over half of the relationship between aerosol optical depth and LCC results from meteorological covariations (Gryspeerdt et al., 2016). Besides, in climate projections, Webb et al. (2012) found that most climate models cannot reproduce the observational LCC-LTS/EIS relationship and thus low cloud feedbacks have the largest spread among climate models. To help constrain future climate projections, the LTS/EIS-induced low cloud feedback can be more accurately estimated by multiplying the observational LCC-LTS/EIS sensitivity by the LTS/EIS changes of climate model projections (Webb et al., 2012; Qu et al., 2014; Myers and Norris, 2016; Klein et al., 2017; Mccoy et al., 2017; Myers et al., 2021; Seethala et al., 2015; Kawai et al., 2017).

Although the LTS/EIS is best correlated with LCC among all meteorological factors, the LTS/EIS only explains a small portion of LCC variance on short time scales. 12% of daily LCC variance are explained by the LTS, but when the monthly means are subtracted from the data only 4.8% of the daily LCC variance are explained by the LTS at the subtropical ocean weather station (OWS) N (Klein, 1997). Similarly, when the monthly means are removed, only 4% of daily LCC variance are explained by the EIS over the typical subtropical eastern oceans (Szoeke et al., 2016). LCC on daily time scales is not as well explained by the LTS/EIS as the LCC on longer time scales. But the LCC sensitivity to LTS/EIS is assumed to be time-scale invariant to estimate the LTS/EIS-induced low cloud feedback and thus leads to some uncertainty (Klein et al., 2017). The explanation for the variant relationship between LCC and LTS/EIS across different time scales is not clear. And it is also not known whether the LTS and EIS can approximate the IS with the same accuracy across different time scales.

Grounded on the well-mixed condition, the PBL's thermal structure is relatively simple and both the LTS and EIS are likely good measures of IS. However, the actual PBL thermal stratification may not always be well-mixed. In deep decoupled PBLs, a strong stratification with a large $\theta$ increase between cloud layers and surface-mixed layers would exist (Jones et al., 2011; Nicholls, 1984). In this case, both the LTS and EIS likely count the stable layer of the decoupling into the IS estimates and thus overestimate the real IS atop the PBL. Previous studies also showed that the free-tropospheric lapse rate has small biases and large spreads although on average it is close to the moist adiabat on daily time scales (Wood and Bretherton, 2006). Thus further refinements on the algorithm of IS estimations are possible if we can reduce the biases and errors resulting from the deviations from the well-mixed conditions. Given the importance of the LTS/EIS for studies of cloud-aerosol interactions and climate predictions, a better measure of the IS can lead to more accurate quantification and increasing confidence in these fields. Based on the previous EIS framework, this study further establishes a profile-based EIS (EIS$_p$) algorithm to take advantages of the reanalysis and thus more accurately estimate the IS.

This paper is laid out as follows: Section 2 briefly describes the observation and reanalysis data and introduces methodologies used in our analysis; section 3 illustrates the development and validation of the new EIS$_p$; section 4 evaluates the relationship between LCC and EIS$_p$ at the global scale; with conclusions in section 5.

## 2. Data and methods

Data used in this study includes: (1) high vertical-resolution radiosondes and cloud radar and lidar observations from the ground site of Atmospheric Radiation Measurement (ARM) Program; (2) radiosondes of several subtropical and midlatitude stations from the Integrated Global Radiosonde Archive (IGRA) of the National Oceanic and Atmospheric Administration (NOAA); (3) global satellite observations of LCC; (4) the fifth-generation atmospheric reanalysis from the European Centre for Medium-Range Weather Forecasts (ECMWF). Methodologies of data processing are also introduced.

## 2.1 Radiosonde and cloud observations at the ground-based sites

Long-term ground-based observations are from two sites of the ARM Program at the Southern Great Plains (SGP) and the Eastern North Atlantic (ENA) (Ackerman and Stokes, 2003). ARM was established by U.S Department of Energy Office of Biological and Environmental Research to provide an observational basis for studying the Earth's climate. At the SGP observatory (97.5ºW, 36.6ºN and 318m above the sea level) and the ENA observatory (28.1ºW, 39.5ºN and 30m above the sea level), high-quality radiosondes and cloud radar and lidar observations are provided to validate the new algorithm of $EIS_p$ and investigate the relationship of IS and IS estimates (i.e., LTS, EIS and $EIS_p$) with LCC. However, the ENA is located on Graciosa Island at the midlatitude ocean where low clouds frequently occur but with no inversion (Norris, 1998) so that it is not an ideal site to investigate the relationship of LCC with IS. Thus, the observations at the ENA are only used to validate the accuracy of $EIS_p$ by comparing with the radiosonde-measured IS.

The atmospheric temperature, relative humidity (RH) and pressure profiles measured by the SGP balloon-borne sounding system (SONDE) from 2002 to 2011 are used. The sondes at the SGP are launched four times a day at 5:30, 11:30, 17:30, 23:30 coordinated universal time (UTC). To avoid the diurnal-cycle influence on our analysis, only the sondes launched at 17:30 UTC (11:00 local time) are used. At this time, the PBL is relatively more well-mixed by turbulence with more uniform vertical distribution of $\theta$ than the other time of a day (Liu and Liang, 2010). The data at different time are also tested and they come to similar results. The precision of the sonde-measured temperature, RH and pressure is 0.1K, 1% and 0.1hPa (Ken, 2001), respectively. Their accuracy is 0.2K, 2% and 0.5hPa, respectively (Ken, 2001). Its vertical resolution is normally about 10 meters from the ground level up to 30km. The sonde temporal resolution is less than 2.5s with 6m/s ascent rate at the 1000hPa level. The $\theta$ profile is computed from the sonde temperature and pressure profiles as:

$$\theta = T(\frac{1000}{p})^{\frac{R_a}{c_{pa}}}, \tag{1}$$

where $R_a$ is the specific gas constant of dry air; $c_{pa}$ is the specific heat capacity for dry air at constant pressures. T and p are the sonde temperature and pressure. The $\theta$ vertical gradient ($d\theta/dz$) profile is derived from the $\theta$ difference between two adjacent levels:

$$(\frac{d\theta}{dz})_{\frac{z_{i+1}+z_i}{2}} = \frac{\theta_{i+1}-\theta_i}{z_{i+1}-z_i}, \tag{2}$$

where $z$ is the height above the ground level (AGL). The subscript "i" indicates the i-th level detected by the sonde.

Cloud profiles are observed every 10s by the 35GHz millimeter wavelength cloud radar and the micro-pulse lidar from 2002 to 2011 at the SGP. The ARM best estimate cloud radiation measurement (armbecldrad) product is used (Chen and Xie, 1996), which provides radar and lidar cloud profiles derived from the Active Remote Sensing of Clouds (ARSCL). Its vertical resolution is 45 meters. To match the sonde launched at 17:30 UTC, the hourly segment of cloud measurements during 17:00-18:00 UTC is used. The cloud base/top height of an hourly segment is recognized as the lowest/highest level of cloud layers (non-zero cloud fraction) detected in that hourly segment. In a cloud profile, distinct cloud layers are separated by a minimum distance threshold of 250m (Li et al., 2011). Low clouds are defined as the cloud base height less than 3km and the top height less than 4km. These low clouds are dominated by stratus, stratocumulus, and shallow cumulus clouds (Dong et al., 2005). Segments of solely other types of clouds but no low cloud are excluded in our analysis. Segments that have low clouds but with other clouds aloft are kept. The LCC of an hourly segment is defined as the ratio of the number of cloudy profiles to the total number of profiles in that segment.

These hourly segments are further sorted into three categories: clear sky, coupled cloudy and decoupled cloudy segments. Clear sky segments are those in which no cloud is present within that segment. The coupled/decoupled cloudy segments are segments containing low-clouds in coupled/decoupled PBLs, respectively. A straightforward indicator to distinguish coupled and decoupled PBLs is the height difference between the cloud base and the LCL ($\Delta z_b$) (Jones et al., 2011). When the PBL is well mixed, $\Delta z_b$ is close to zero, but in the decoupled PBLs the cloud and subcloud layers would be separated by a stable layer

and the LCL may diverge from the cloud base hundreds of meters with large $\Delta z_b$ (Nicholls, 1984; Jones et al., 2011). The threshold value of $\Delta z_b$ is empirical and for different instrument capability, vertical resolution and locations the threshold may be a little different. In reference to the linear least-square fit between $\Delta z_b$ and $\Delta \theta$ in Jones et al. (2011) that 150 meters of $\Delta z_b$ correspond to 0.5K of the $\theta$ difference in the subcloud layer, a similar linear relationship is found but the slope is a little different that 180 meters of $\Delta z_b$ corresponds to 0.5K of the $\theta$ difference at the SGP site. Thus at the SGP site, a threshold value of 180 meters for $\Delta z_b$ is used to distinguish coupled and decoupled PBLs.

At the ENA, data of radiosondes and LCC from 2014 to 2020 are used. The data product and processing method of the ENA site is the same to that of the SGP. The ENA site is characterized by marine stratocumulus clouds but at midlatitude where the correlation between LCC and IS is much weaker as compared to that at the SGP. This will be verified and discussed later.

**2.2 Radiosonde stations of subtropics and midlatitude**

The IGRA of NOAA collects radiosondes from global distributed stations (Durre et al., 2018; Durre et al., 2006). The radiosonde temperature, RH, pressure, and geopotential height profiles in the IGRA are used. The $\theta$ and $\theta$ gradient profiles are computed from Eqs. (1) and (2). These atmospheric parameters of radiosondes are available at the standard pressure levels (1000, 925, 850, 700 and 500hPa) or variable levels. It provides reliable instantaneous observations for the PBL IS (see definitions in the section 2.5). However, most low-cloud dominated regions are over the ocean with no available radiosondes in the IGRA. Thus five radiosonde stations with relatively higher occurrence frequencies of low clouds are selected: the OWS N in the subsidence and steady trade wind circulation of the northeast Pacific (Klein, 1997; Klein et al., 1995); the OWS C in the frequently decoupled PBLs of the north Atlantic (Norris, 1998); the tropical east Pacific coast with the classic stratocumulus condition (Albrecht et al., 1995); the southeast Pacific coast with the stratocumulus-capped PBLs (Bretherton et al., 2004) and the southeast Chinese coast of subtropical low-cloud domains (Klein and Hartmann, 1993). Locations, observational period and time of data for each station are listed in Table1.

Table 1. The location, observational period and time of the IGRA radiosonde stations.

| | OWS N | OWS C | Tropical East Pacific coast | Southeast Pacific coast | Chinese coast |
|---|---|---|---|---|---|
| Location | (140ºW, 30ºN) | (35.5ºW, 52.75ºN) | (120.5667ºW, 34.75ºN) | (70.4408ºW, 23.4503ºS) | (119.2833ºE, 26.0833ºN) |
| Period | 1969-1974 | 1969-1974 | 2006-2011 | 2006-2011 | 2006-2011 |
| Time | 00UTC | 12UTC | 00UTC | 12UTC | 00UTC |

**2.3 Global LCC observations**

Global LCC observations of the geostationary satellites (GEOs) and the Moderate Resolution Imaging Spectroradiometer (MODIS) onboard the Aqua and Terra satellites are provided by the Clouds and the Earth's Radiant Energy System (CERES) project (Doelling et al., 2013; Doelling et al., 2016; Trepte et al., 2019). Global hourly LCC between 60ºS and 60ºN during 2006-2011 is used. It is available in the synoptic 1-degree (SYN1deg) edition 4.1 product of the CERES project (Doelling et al., 2013; Doelling et al., 2016; Trepte et al., 2019). The GEO-MODIS LCC here refers to the cloud area fraction of the identified cloudy pixels with cloud top pressure above 700hPa divided by the total number of pixels in the 1º×1º grids. The MODIS pixel-level cloud identification is based on the CERES MODIS cloud algorithm (Minnis et al., 2008; Minnis et al., 2011). The sampling frequency of clouds derived from the MODIS narrowband radiance is four times a day (two from each of the Aqua Terra). GEOs with radiances calibrated against the MODIS provide hourly cloud retrievals between MODIS observations (Doelling et al., 2013). The GEO cloudy pixel identification is also based on the CERES MODIS-like cloud algorithm to achieve more uniform MODIS and GEO clouds. An advantage of this product over cloud retrievals of the first-

generation GEO is that the CERES project uses the latest generation of the GEO imager capability with more additional channels to enhance the accuracy of cloud retrievals (Doelling et al., 2016). Hourly LCC is used to match the IGRA radiosondes. Daily LCC used in section 4 is the mean of the full-day hourly GEO-MODIS LCC from the CERES SYN1deg Ed4.1 product (Doelling et al., 2016).

**2.4 The fifth generation ECMWF atmospheric reanalysis (ERA5)**

Reanalysis data from the ECMWF is to provide the atmospheric profile information. The ERA5 combines observations with model outputs by the 4D-Var assimilation to achieve the 1-hour resolution (Hersbach et al., 2020). The hourly atmospheric temperature, RH, geopotential profiles in the ERA5 dataset are used to match the SGP, IGRA and GEO-MODIS observations. The $\theta$ and $\theta$ gradient profiles are computed based on Eqs. (1) and (2). Atmospheric profiles at the 16 pressure levels between 500hPa and 1000hPa are available. At the SGP site, the ERA5 atmospheric profiles between the years 2002 and 2011 at the grid point (97.5ºW, 36.625ºN) nearest to the SGP site (within about 2.8km) is used. For the IGRA radiosonde stations, the ERA5 hourly data of the 0.125º grid point nearest to them during the same observational period is used. At the global scale, the ERA5 atmospheric profiles are averaged to 1º resolution data centered at 0.5º, 1.5º, … during the years between 2006 and 2011. This resolution is consistent with the global LCC data. Those three metrics, LTS, EIS and $EIS_p$, are then computed based on the 3-hour 1º ERA5 atmospheric profiles. All metrics at longer (i.e., from daily to seasonal) time scales are computed from the 3-hour metrics.

**2.5 LTS, EIS and radiosonde-measured IS**

The LTS and EIS over the ocean are defined as:

$$\text{LTS} = \theta_{700hPa} - \theta_0, \tag{3}$$

$$\text{EIS} = \text{LTS} - \Gamma_m \left( z_{700hPa} - z_{LCL} \right), \tag{4}$$

where $\theta$ and $z$ are, respectively, the potential temperature and the height. The subscripts "700hPa", "0" and "LCL" indicate the levels of 700hPa, 1000hPa and the LCL, respectively. $z_{LCL}$ is calculated using temperature and RH at 1000hPa based on the exact expression in Romps (2017), indicating the height at which an air parcel would saturate if lifted adiabatically. $\Gamma_m$ is the moist-adiabatic $\theta$ gradient at 850hPa calculated using the mean temperature of the 1000hPa and 700hPa levels. $\Gamma_m$ can be calculated as:

$$\Gamma_m\left(T,p\right) = \left(\frac{1000}{p}\right)^{\frac{R_a}{c_{pa}}} \cdot \frac{g}{c_{pa}} \left(1 - \frac{1 + L_v q_s(T,p)/R_a T}{1 + L_v^2 q_s(T,p)/c_{pa} R_v T^2}\right). \tag{5}$$

$q_s$ is the saturated mass fraction of water vapor. $L_v$ is the latent heat of vaporization. $R_v$ is the specific gas constant for water vapor.

Over land, the LTS and EIS are computed following Eqs. (3)-(5) but based on the heights of 0.15km and 3km AGL. The height of the initial air parcel set as 0.15km AGL is to avoid noisy and contaminated readings of the RH near the surface from the radiosondes and the influence of surface layers (Liu and Liang, 2010). The temperature, RH and pressure at 0.15km and 3km AGL over land can be directly derived from the radiosondes or linearly interpolated from the ERA5 profiles. $z_{LCL}$ over land is calculated using the temperature and RH at 0.15km AGL. $\Gamma_m$ over land is computed using the mean temperature and pressure of the two heights.

To derive the IS from the radiosonde profiles, the layer of the greatest $\theta$ gradient ($d\theta/dz$) between the LCL and 5km AGL is firstly identified, similar to Mohrmann et al. (2019) but with a LCL constraint to guarantee that it is above the cloud layer. For the SGP high-resolution (10 meters) radiosondes, the inversion top/base is defined as the nearest level above/below the layer of maximum $d\theta/dz$ where $d\theta/dz$ equals to three-fourths of maximum $d\theta/dz$. The IS is defined as the $\theta$ jump across the inversion layer after removing the $\theta$ increase due to the moist adiabat in this layer:

$$\text{IS} = \left(\theta_{IST} - \theta_{ISB}\right) - \Gamma_m^{ISB}(z_{IST} - z_{ISB}). \tag{6}$$

The subscripts "IST" and "ISB" indicate the identified top and base height of corresponding layers, respectively. $\Gamma_m^{ISB}$ is the moist-adiabatic $d\theta/dz$ computed from Eq. (5) using the temperature and pressure at the identified inversion base. The method that determines the IS in low-resolution soundings of IGRA is exactly the same as the new profile-based method of EIS and will be introduced in detail in section 3.1.

**2.6 t-test and multiple timescale analysis**

In our study the Pearson's correlation coefficient (R) and the slope of the least-squares linear fit are used. R-square is used with a minus/plus sign for a negative/positive correlation. The existence of a correlation and confidence interval for the true mean value ($\mu$) are estimated based on the t-test. The number of independent samples is determined by dividing the total length of samples by the distance between independent samples (Bretherton et al., 1999). All correlations listed in this study are at the 95% significant level if without a mention of their significance. The confidence bound of R is computed based on the Fisher-Z Transformation. The confidence interval of the slope is computed from the residual error of the least-squares linear fit. Besides, for isolating the correlation and the regression slope on different time scales, window anomalies are defined as consistent with that in Szoeke et al. (2016):

$$x^{\Delta_i} = [x]^{\Delta_i} - [x]^{\Delta_{i+1}}. \tag{7}$$

The brackets represent mean of $x$ over the window of length $\Delta$. The superscripts $\Delta_i$ and $\Delta_{i+1}$ are the i-th window length and the next longer window length.

**3. The profile-based method of EIS (EIS$_p$)**

In this section, the new EIS$_p$ algorithm is established based on ground-based observations at the SGP and validated at other radiosonde stations of subtropics and midaltitude. In section 3.1, the new EIS$_p$ algorithm is described. In section 3.2, at the SGP site with long-term 10m-resolution radiosondes, two questions are discussed: (1) why and how is EIS$_p$ a better estimate for the IS than LTS and EIS? (2) how well does EIS$_p$ control LCC as compared to LTS and EIS when it is a better estimate for the IS? In section 3.3, the EIS$_p$ is further validated at radiosonde stations of subtropics and midlatitude.

**3.1 The algorithm of the new EIS$_p$**

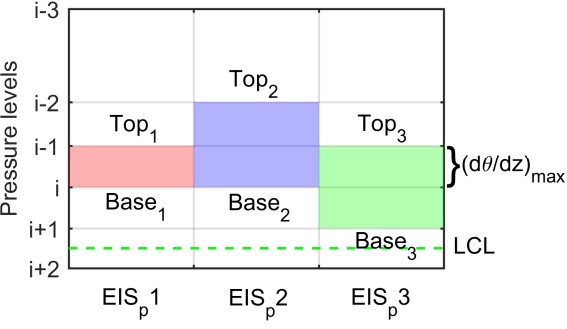

Figure 1. An illustration of finding the location of three possible layers encompassing the inversion between the LCL and 5km AGL in ERA5 or coarse sounding profiles. The red block is one single layer of $(d\theta/dz)_{max}$ that includes the inversion. The blue and green blocks are a combination of two adjacent layers if the inversion is distributed into the two layers but not just in the layer of $(d\theta/dz)_{max}$. EIS$_p$1-3 are computed accordingly and the largest value among them is regarded as the true EIS$_p$.

The EIS$_p$ is designed to capture the IS information from the thinnest layer encompassing the inversion in low-resolution (hundreds of meters) atmospheric profiles. For these coarse-resolution profiles (e.g., ERA5), it is difficult to accurately locate the exact place of the inversion because usually the thickness of the inversion is much smaller than the distance between two

adjacent vertical levels. Thus only one or two adjacent layers that could encompass the inversion are located. The latter is for the consideration that an inversion layer may be across two adjacent layers of the ERA5. Specifically, the $EIS_p$ is computed as follows.

(1) *Locating the layer of the maximum $\theta$ vertical gradient $(d\theta/dz)_{max}$:*

For each hourly ERA5 profile, the layer of $(d\theta/dz)_{max}$ is firstly located between the LCL and 5km AGL (the red zone in Fig.1), since the inversion just features strong gradients in thermodynamical properties.

(2) *Finding the layers encompassing the full inversion:*

The layer of $(d\theta/dz)_{max}$ may not encompass the full inversion if the inversion crosses two adjacent layers of the ERA5. Thus, the layer of $(d\theta/dz)_{max}$ is combined with an adjacent layer just above and below it respectively, to constitute other two candidate layers that could encompass the full inversion (the blue and green zone in Fig.1).

(3) *Calculating the $EIS_p$:*

The $EIS_p$ is calculated for the three possible layers identified in second stage, respectively:

$$EIS_p = \theta_{top} - \theta_{base} - \Gamma_m(z_{top} - z_{base}), \tag{8}$$

where subscripts "top" and "base" represent the top and base levels of a candidate layer. $\Gamma_m$ is computed using Eq. (5) at the base level. The $\theta$ increase of the moist adiabat is removed to extract the strength of the inversion between the top and base levels, which is consistent with the EIS framework in Wood and Bretherton (2006). The final $EIS_p$ is determined by which layer in Fig.1 encompasses stronger inversion computed from Eq. (8) and thus refers to the largest value among the three candidates $EIS_p$1-3.

The EIS (Wood and Bretherton, 2006) assumes that the PBL is well mixed (dry adiabat below the LCL and moist adiabat above the LCL) for estimating the IS. If that is the case, $EIS_p$ would give the same results as EIS. However, it will be shown in the following sections that the actual PBL often deviates from the well mixed conditions, where the $EIS_p$ provides a physically more reasonable estimate for the IS than the EIS and thus a stronger cloud-controlling factor.

When high-resolution radiosondes are available, the exact IS can be obtained fairly straightforward (section 2.5, Eq. 6). The computation of $EIS_p$ is in fact adapted from the algorithm of obtaining the IS from high-resolution radiosondes, but is adjusted to suit coarse-resolution atmospheric profiles in reanalysis. Because high-resolution soundings are rare, an applicable metric derived from reanalysis would be much more beneficial. Because the IGRA soundings have similar vertical resolutions as ERA5 in lower troposphere, the IS of these soundings (used in section 3.3) is derived exactly by the same way as the $EIS_p$.

## 3.2 PBL stratification and the establishment of the $EIS_p$ at the SGP

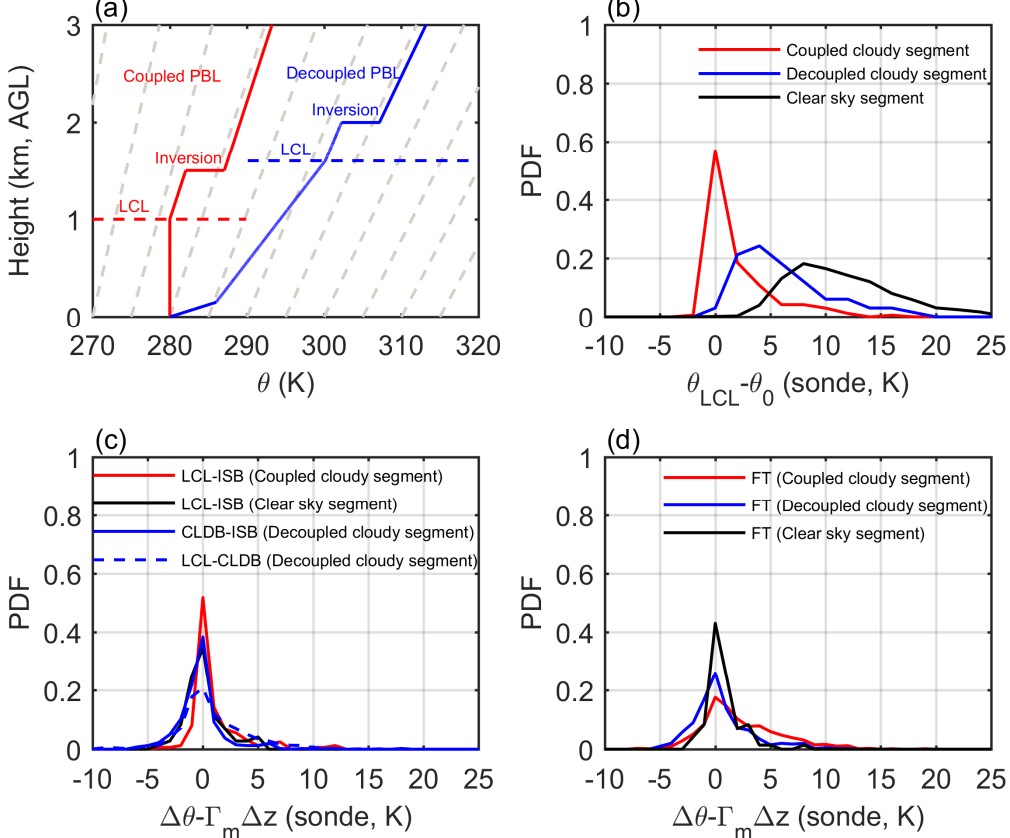

Figure 2. Illustrations of PBL $\theta$ profiles (a), with the LCL heights indicated by horizontal dash lines and the moist adiabat represented by light dash lines. PDFs of the $\theta$ difference between the LCL and 150m AGL (b), the $\theta$ difference with the moist adiabat removed between the LCL and the inversion base (c) and the $\theta$ difference with the moist adiabat removed for the free troposphere between the inversion top and 3km AGL (d). The red, blue and black lines are for coupled cloudy, decoupled cloudy and clear sky segments, respectively. In (c), the $\theta$ differences of decoupled cloudy segments are further separated into that between the LCL and the cloud base (blue dash line) and that between the cloud base and the inversion base (blue solid line).

The characteristics of PBL thermal structures are examined by using the SGP high-resolution radiosondes as shown in Fig.2. Fig. 2a illustrates an idealized $\theta$ profile of the well-mixed condition consistent with Wood and Bretherton (2006) and an idealized $\theta$ profile of the decoupled PBLs based on the observations in Jones et al. (2011). The primary difference in the $\theta$ profiles between the coupled and decoupled PBLs is whether a stable layer exists to decouple the cloud and subcloud layers (Nicholls, 1984). Hence, under the decoupled conditions, the LTS and EIS would include the sum of the PBL IS and the $\theta$ increase from the ground to the LCL (the blue line in Fig. 2a). The LTS and EIS can be separated into different terms:

$$\text{LTS} = (\theta_{LCL} - \theta_0) + \Delta\theta + \text{IS}, \tag{9a}$$

$$\text{EIS} = (\theta_{LCL} - \theta_0) + (\Delta\theta - \Gamma_m \Delta z) + \text{IS}, \tag{9b}$$

$$\Delta\theta = \theta_{3km} - \theta_{LCL} - \text{IS}, \tag{9c}$$

$$\Delta z = z_{3km} - z_{LCL}. \tag{9d}$$

The subscripts of "3km", "0" and "LCL" indicate the levels of 3km, 150m AGL and LCL. If over oceans, the levels of 3km and 150m can be replaced with 700hPa and 1000hPa. In Eq. (9a), the LTS can be regarded as the sum of the $\theta$ difference between the LCL and 150m AGL ($\theta_{LCL} - \theta_0$), the $\theta$ increase ($\Delta\theta$) due to the actual $\theta$ gradient above the LCL, and the PBL IS. Similarly in Eq. (9b), the EIS is similar to the LTS except that the $\theta$ increase due to the moist adiabat ($\Gamma_m \Delta z$) above the LCL is removed. It can be seen that the first two terms on the rhs of Eqs. (9a) and (9b) are contributing to the LTS and EIS even though they are not a part of the IS. In the well-mixed PBLs, the two terms $\theta_{LCL} - \theta_0$ and $\Delta\theta - \Gamma_m \Delta z$ are both equal to zero. Thus the EIS defined as Eq. (9b) is exactly the IS and the LTS defined as Eq. (9a) equals to $\text{IS} + \Gamma_m \Delta z$ under perfectly

well-mixed conditions.

At the SGP site, 29%, 32% and 39% observational samples are classified into the coupled cloudy, decoupled cloudy and clear sky segments, respectively. Note that the $\Delta z_b$ method cannot distinguish whether the PBL is coupled or decoupled when a segment has no low cloud. Thus the clear sky segments might contain both coupled and decoupled PBL. In Fig. 2b: a) the probability distribution functions (PDFs) of $\theta_{LCL} - \theta_0$ for the coupled cloudy segments peak at zero and relatively have positive skewness. The exact reason of the positive skewness is not clear. Because the height of LCL being close to the simultaneously observed cloud base height is only a necessary condition of a PBL being coupled. A decoupled surface layer and overlaying cloud layer coincidently have the height of LCL close to the cloud base is not a surprise. Either clouds advected from other places or a new surface stable layer has developed while clouds formed earlier are still left above might result in positive $\theta_{LCL} - \theta_0$. b) Strong stratification below the LCL (large positive $\theta_{LCL} - \theta_0$) frequently occurs in the decoupled cloudy and clear sky segments with mean value of 6.3K and 11.5K, respectively. Thus the non-zero term of $\theta_{LCL} - \theta_0$ will cause LTS and EIS to largely deviate from the real value of IS in the decoupled cloudy and clear sky segments.

Besides, a premise of using LTS and EIS to measure the IS is that the lower-tropospheric $\theta$ gradient can be predicted by the moist adiabat above the LCL. This moist adiabatic assumption is supported in previous studies but still with some uncertainties on the daily time scales (Stone, 1972; Wood and Bretherton, 2006; Schneider and O'gorman, 2008). According to PDFs of the $\theta$ difference between the LCL and inversion base or between the inversion top and 3km AGL with the moist adiabat removed ($\Delta\theta - \Gamma_m\Delta z$), $\theta$ likely follows the moist adiabat above the LCL (Figs. 2c and 2d) with a peak at zero but all PDFs of $\Delta\theta - \Gamma_m\Delta z$ have broad distributions. The standard deviation of $\Delta\theta - \Gamma_m\Delta z$ above the LCL is about 4K. Note that here the $\Gamma_m$ is computed using the Eq. (5) but based on the temperature and pressure at the base level of each layer.

Typically, the real IS is less than 10K. Thus the term $\theta_{LCL} - \theta_0$ in Eqs. (9a) and (9b) will cause a strong overestimate of the IS by the LTS and EIS. And the variation of the LTS and EIS is attributed to not just variations of IS but also variations of the systematical deviations of temperature profiles from the dry adiabat below the LCL. As a result, at the SGP site, the decoupled cloudy and clear-sky segments (with weak IS but large $\theta_{LCL} - \theta_0$) are mixed with the coupled cloudy segments with strong IS when using the LTS and EIS to sort data. Large values of LTS and EIS correspond to not just strong IS but also weak IS with strong stratification below the LCL. On short time scales (like the daily scale), the spread of $\Delta\theta - \Gamma_m\Delta z$ (Figs. 2c and 2d) resulting from the $\theta$ gradient deviating from the moist adiabat above the LCL could add additional uncertainty into the LTS/EIS. Hence, weak and even unphysical relationships of clouds and moisture with the LTS/EIS might exist.

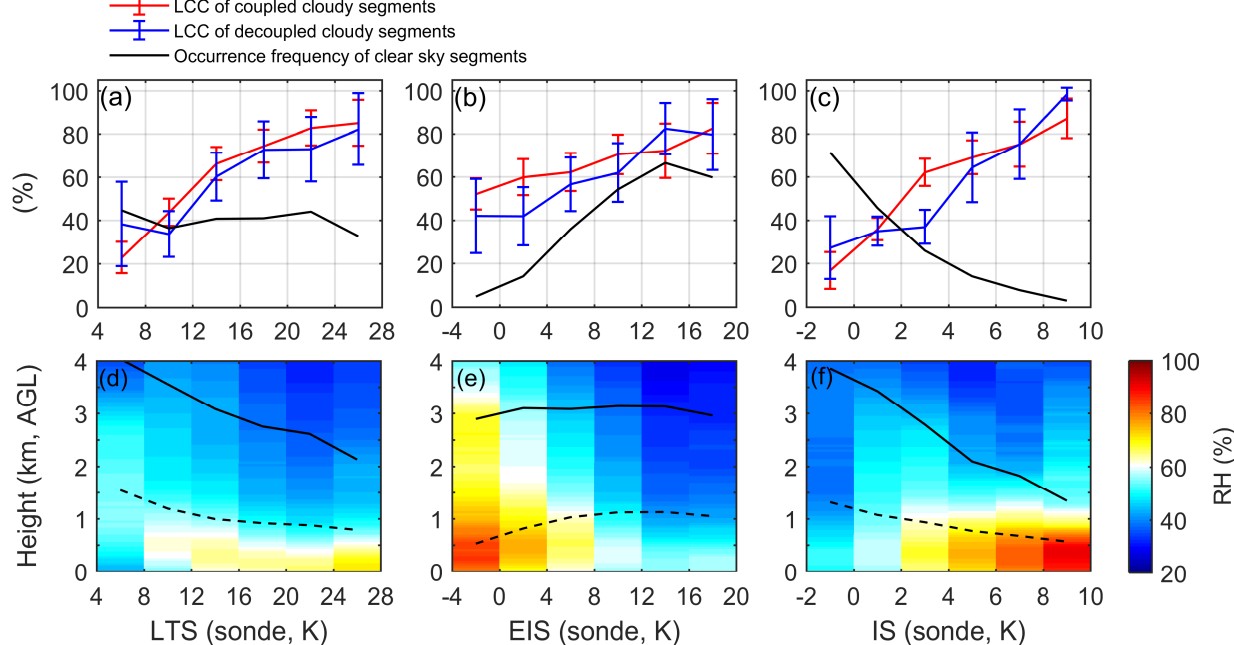

Figure 3. Top panel: LCC composites of the coupled cloudy (red line) and decoupled cloudy segments (blue line) and the occurrence frequency of the clear sky segments (black line). Bottom panel: composited RH profiles. Composites are based on the SGP radiosonde-measured LTS (a and d), EIS (b and e) and IS (c and f), respectively. Error bars in (a), (b) and (c) show the 95% confidence interval of the mean based on the t-test. The solid and dash black lines in (d), (e) and (f) indicate the average height of the inversion center and the LCL, respectively. All composites are based on daily data of all seasons for the full period at the SGP.

Figs. 3a-c show that the composited LCC of cloudy segments are all positively proportional to the radiosonde-measured LTS, EIS and IS. However, the composites of LCC are slightly/significantly more sensitive to the changes of IS than the LTS/EIS. The occurrence frequency of the clear sky segments (the number of clear-sky segments divided by the number of total segments) is investigated separately. Fig. 3c shows that clear sky segments are rarely observed when the IS is very strong (~0% at 10K), and more frequently exist towards weaker IS (60% at 0K). This is consistent with that stronger IS inhibits the entrainment of dry air from the free troposphere and thus favors the formation and maintenance of low clouds and corresponds to less occurrence of the clear sky. On the contrary, such a physically reasonable expectation is not seen (even qualitatively) in the composites of the clear sky segments based on the LTS and EIS. Figs. 3a-b show that the occurrence frequency of clear sky segments changes little (even increases) with increasing LTS (EIS). This is also expected based on Fig. 2b showing the existence of a large positive skewness in the term $\theta_{LCL} - \theta_0$ in the clear sky segments. This strong static stability below the LCL results in large LTS and EIS even when the real IS is weak.

Composited moisture distribution shows consistent information with the LCC composites. Fig. 3f shows that the composited RH has an increasing trend towards stronger IS and high values of RH (RH>80%) are restricted below 1km AGL at the large IS value bins. However, the composited RH distribution is completely reversed when sorted by the EIS, with high/low RH related to weak/strong EIS (Fig. 3d). The RH distribution sorted by the LTS has similar dependence on the magnitude of the LTS (Fig. 3c) to the IS, but with weaker variations and smaller PBL RH as compared to the composites based on the IS (Fig. 3e). Thus the LTS/EIS poorly/incorrectly represents the IS at the SGP site, and hence the dependence of the PBL moisture conditions and LCC on the IS are weakly/erroneously reproduced by the LTS/EIS.

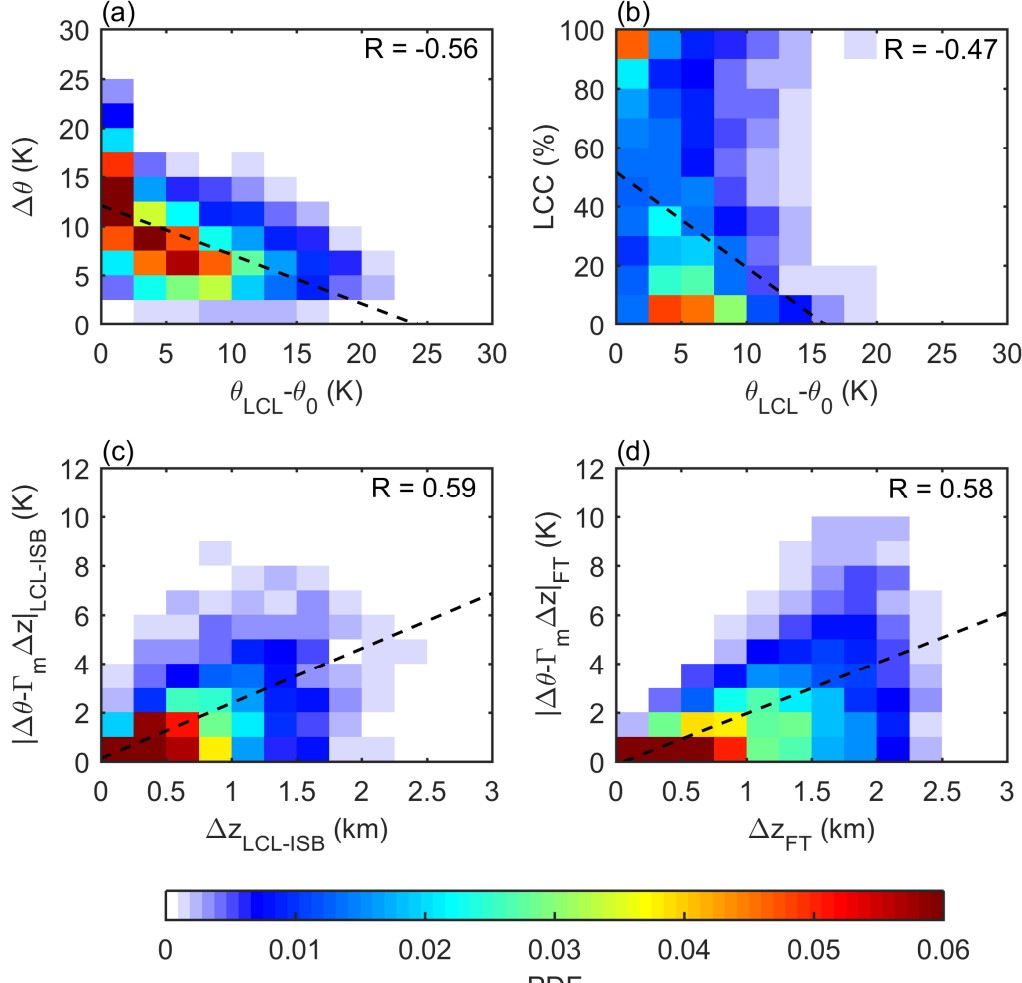

Figure 4. Joint PDFs of the $\theta$ difference ($\Delta\theta$) between the levels of 3km and the LCL (with the IS excluded) and $\theta_{LCL} - \theta_0$ (a), and PDFs of LCC and $\theta_{LCL} - \theta_0$ (b). Joint PDFs of the absolute value of the $\theta$ difference with the moist adiabat removed ($\Delta\theta - \Gamma_m\Delta z$) and the height difference ($\Delta z$) from the LCL to the inversion base (c) and from the inversion top to 3km in the free troposphere (d). Correlation coefficients (R) are listed on the upper-right corner of each panel. The black dash lines indicate the least-squares fit.

An interesting phenomenon is that the LTS overall performs better than the EIS with respect to constraining LCC at the SGP site. To understand why this happens, the LTS and EIS in Eq. (9) both have been separated into three terms to discuss. For the LTS, the two terms $\theta_{LCL} - \theta_0$ and $\Delta\theta$ of Eq. (9a) usually offset each other with a negative correlation of -0.56 and a slope of the least-squares fit -0.5K/K (Fig. 4a). In contrast, the slope of the least-squares fit between $\Delta\theta - \Gamma_m\Delta z$ and $\theta_{LCL} - \theta_0$ is only -0.05K/K (not shown). Furthermore, the LTS and EIS equation can be transformed into:

$$LTS = \left(1 + \frac{\Delta\theta}{\theta_{LCL} - \theta_0}\right)(\theta_{LCL} - \theta_0) + IS, \tag{10a}$$

$$EIS = \left(1 + \frac{\Delta\theta - \Gamma_m\Delta z}{\theta_{LCL} - \theta_0}\right)(\theta_{LCL} - \theta_0) + IS. \tag{10b}$$

On average, the coefficient before $\theta_{LCL} - \theta_0$ for the LTS in Eq. (10a) is 0.5 while that for EIS in Eq. (10b) is 0.95. The variation of LTS and EIS result from both the changes of IS (positively correlated with LCC as shown in Fig. 3c) and the changes of $\theta_{LCL} - \theta_0$ (negatively correlated with LCC as shown in Fig. 4b). According to Eqs. (10a) and (10b), the LTS actually only involves half of the bias caused by $\theta_{LCL} - \theta_0$ and thus not as strongly influenced by $\theta_{LCL} - \theta_0$ as the EIS. As a result, only removing the moist adiabat ($\Gamma_m\Delta z$) does not make the EIS a better estimate for the IS at the SGP but make the EIS more influenced by $\theta_{LCL} - \theta_0$. This explains why the LTS is better correlated with LCC and RH (Figs. 3a and 3d) than the EIS (Figs. 3b and 3e) at the SGP. However, the physical reason that why the PBL stratification changes in this way is unclear to us and it is beyond the scope of this study.

As shown in Figs. 2c and 2d, the $\theta$ difference between the actual environmental $\theta$ gradient and the moist adiabatic $\theta$ gradient ($\Delta\theta - \Gamma_m\Delta z$) is another source of uncertainty in the EIS based on Eq. (9b), especially on short time scales. However, Figs. 4c and 4d suggest that the spread of $|\Delta\theta - \Gamma_m\Delta z|$ increases with the layer thickness either between the LCL and the inversion base or between the inversion top and 3km AGL (with a correlation of 0.59 or 0.58, respectively). Thus, the thicker the layer encompassing inversion involved in the EIS calculation is, the larger the uncertainty is. Including more layers around the inversion layer in estimating the IS likely results in more uncertainty. This suggests a possible way of better estimating the IS if we can reduce the layer thickness ($\Delta z$) associated with the second term on the rhs of Eq. (9b), which also makes the IS estimate less dependent on the moist adiabatic assumption.

Above results suggest that there are two major bias and error sources of estimating the IS using the LTS and EIS metrics. One is caused by systematic deviations from the dry adiabat below the LCL, the other is the errors resulting from the spread of the actual $\theta$ gradient around the moist adiabat above the LCL. To exclude the former source, we can locate the LCL and only consider the inversion above the LCL to drop the first term on the rhs of Eq. (9b). The impact of the latter one can be indirectly reduced by finding the thinnest layer encompassing the inversion that is involved in the computation of the second term on the rhs of Eq. (9b). Thus, the new EIS$_p$ (as described in section 3.1) is proposed accordingly to achieve a better estimate of the IS.

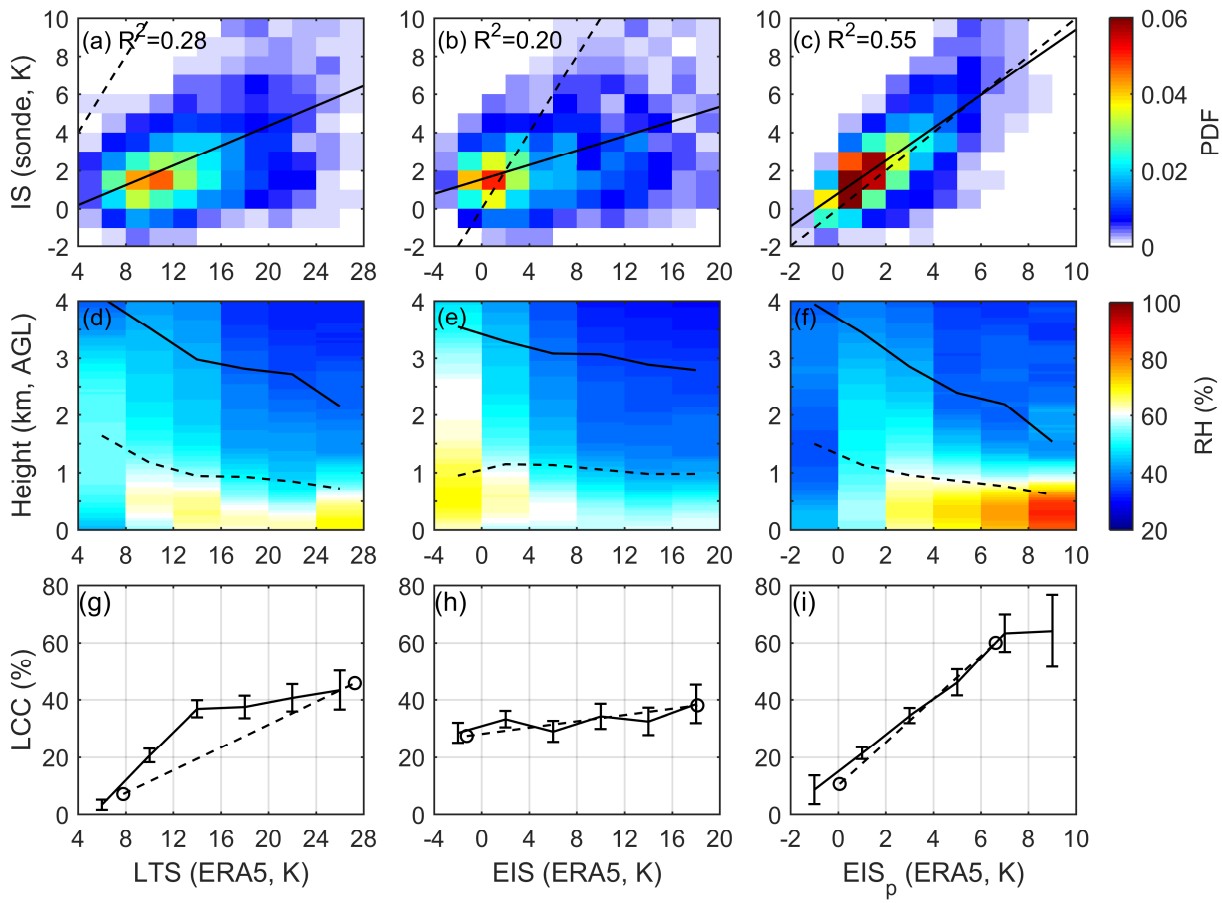

Figure 5. Joint PDFs of the SGP radiosonde-measured IS, and the ERA5-derived LTS (a), EIS (b) and EIS$_p$ (c), respectively. In (a)-(c), the black solid line is the least-squares fit and the dash line is the reference line of y=x. The composites of the radiosonde RH profiles based on the ERA5-derived LTS (d), EIS (e) and EIS$_p$ (f). The black solid and dash lines in (d-f) are the heights of the IS and the LCL, respectively. The LCC composited based on the LTS, EIS and EIS$_p$ are shown in (g), (h) and (i), respectively. The cycles in (g), (h) and (i) corresponds to the 5% and 95% quantile of LTS, EIS and EIS$_p$ and the composited value of LCC in the bins of the smallest and largest 10% of LTS, EIS and EIS$_p$ values. Error bars in (g-i) show the 95% confidence interval of the mean based on the t-test.

The LTS, EIS and EIS$_p$ derived from the hourly ERA5 reanalysis are directly compared against the SGP radiosonde-measured IS. In Fig. 5c, the R-square between the EIS$_p$ estimated from the ERA5 and the IS measured by radiosondes is 0.55,

which is much larger than that of the LTS (0.28, Fig. 5a) and EIS (0.20, Fig. 5b). The slope of the least-squares fit of the IS to the $EIS_p$ is 0.86K/K. This indicates the value of the $EIS_p$ is much closer to the IS as compared to the LTS (0.26K/K) and EIS (0.19K/K). The composites of LCC and RH based on the $EIS_p$ (Fig. 5f) show similar results to that based on the IS (Fig. 3f). Stronger $EIS_p$ corresponds to larger RH trapped below about 1km, and with the $EIS_p$ weakening and the inversion layer lifting RH decreases but distributes to higher levels. However, the LCC and RH composites based on the LTS and EIS (Figs. 5d, 5e, 5g and 5h) show weak or erroneous relationships similar to the results based on the radiosonde-measured LTS and EIS (Fig. 3a, 3b, 3d and 3e). Thus the $EIS_p$ offers a better fit to the real IS and better constrains the PBL moisture distribution and LCC. The slope of the composited LCC to the $EIS_p$ is 6%/K, in contrast to that to the LTS (1.9%/K) and the EIS (0.4%/K). Since the range of the LTS and EIS is larger than that of the $EIS_p$, larger slopes of the LCC to the $EIS_p$ than that to the LTS and EIS are expected. To measure the sensitivity of LCC to changes of LTS, EIS and $EIS_p$, we consider the effective range of LCC resolved by changes in a metric. The sensitivity of LCC to a metric here is defined as the difference between the composited LCC values associated with the largest and smallest 10% of that metric:

$$LCC\ Sensitivity\ to\ x = \overline{LCC(x \geq x_{90\%})} - \overline{LCC(x \leq x_{10\%})}. \tag{11}$$

The bar over the LCC head represents the mean value of LCC sorted by $x$ quantile. $x_{90\%}$ and $x_{10\%}$ are 90% and 10% quantile of $x$. The LCC sensitivity of all segments to the $EIS_p$ is 50%, which is larger than the LTS (39%) and EIS (12%). These weaker/erroneous dependences of LCC on the LTS/EIS are expected since large errors (Figs. 2b-2d) are carried in the LTS/EIS. Although the vertical resolution of the ERA5 profiles may not always suffice to resolve the inversion layer, the IS estimated from the ERA5 profile-based algorithm ($EIS_p$) is highly consistent with the IS directly derived from the SGP 10m-resolution radiosondes and they present similar relationships with the PBL RH and LCC.

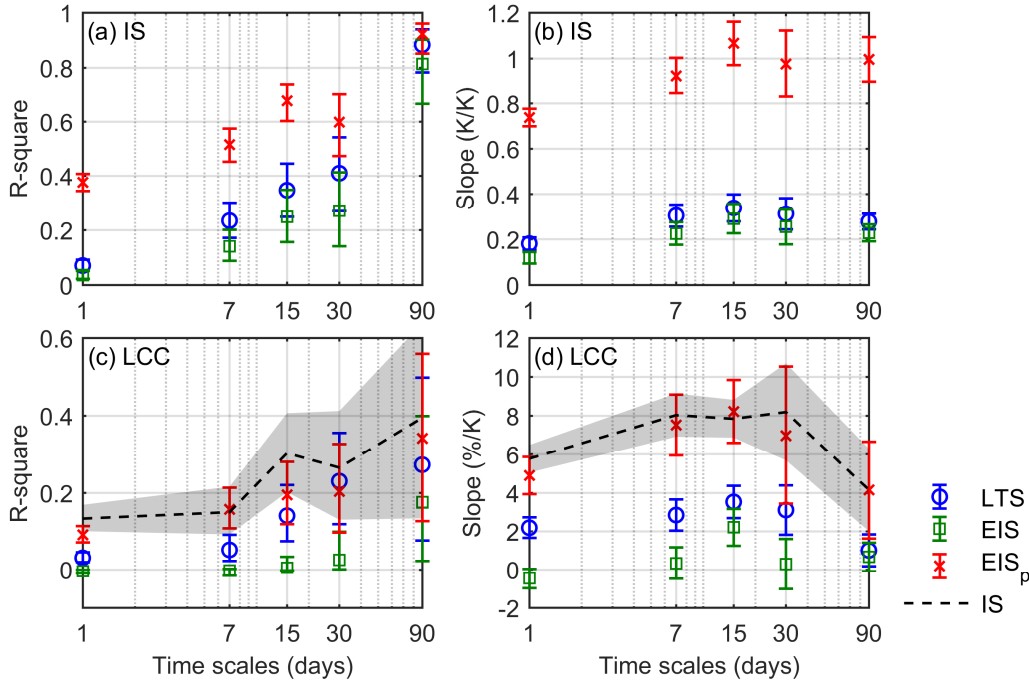

Figure 6. R-square (a) and slope of the least-squares fit (b) of the SGP radiosonde-derived IS to the ERA5 reanalysis-based LTS (blue cycle), EIS (green square) and $EIS_p$ (red cross) on daily, 7-day, 15-day, 30-day and 90-day time scales, respectively. R-square (c) and slope (d) of LCC to the LTS, EIS, $EIS_p$ and IS (black dash line) on daily to seasonal time scales, respectively. Error bars and shadows show the 95% confidence interval of the mean based on the t-test.

The ERA5-based LTS, EIS and $EIS_p$ are further examined on the different time scales with respect to their relationships with radiosonde-measured IS and LCC (Fig. 6). Overall, the R-square and the slope of the $EIS_p$ with the IS are the largest through all time scales as compared to that of the LTS and EIS. Particularly on the daily, 7-day and 15-day time scales, the lower bounds of the 95% confidence interval of the $EIS_p$-IS R-square are much higher than the upper bounds for the LTS and

EIS. On the seasonal time scale, three metrics have similar correlations with the IS, but as shown in Fig. 6b, the slope of the IS to the $EIS_p$ (nearly 1) is still much larger than that to the LTS (0.28K/K) and EIS (0.23K/K). The limited accuracy restricts the LTS and EIS to reproduce the relationship between the true IS and LCC. In Fig. 6c, on daily time scales, the LTS explains 3.1% of variance in LCC, which is comparable to the 4.8% explained variance by the LTS at OWS N (a typical low-cloud dominated site over the ocean) in Klein (1997). For the $EIS_p$, it explains 9.1% of the daily LCC variance, which is remarkably close to that explained by the IS. Similar conclusions can be drawn from weekly time scales. On longer time scales, the $EIS_p$ and the LTS both explain comparable variance in LCC but much larger than that explained by the EIS. In Fig. 6d, the slope of LCC composited based on the IS is nearly reproduced by the $EIS_p$ consistently. The slopes of LCC composited based on the LTS and EIS are much smaller than that based on the $EIS_p$ and IS.

### 3.3 Validation of the $EIS_p$ at radiosonde stations of subtropics and midlatitude

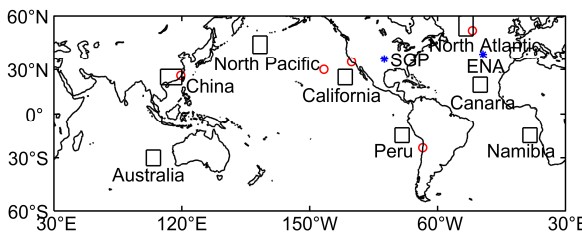

Figure 7. Blue asterisk marks the SGP and ENA sites. Red cycles mark the locations of radiosonde stations from the IGRA. Eight 10º×10º boxes are the most typical low-cloud dominated regions defined in Klein and Hartmann (1993).

Table 2. The characterestics of the PBL thermal strucures, evaluation of the LTS, EIS and $EIS_p$ on estimating the IS and the IS-LCC relationships of the six radiosonde stations. Coupled and decoupled PBLs of all stations are distinguished by $\alpha_\theta$. Italic indicates not significant correlations. Bold indicates the largest correlation. The daily IS-LCC correlation is based on the data after subtracting 7-day means.

| | ARM SGP | ARM ENA | OWS N | OWS C | Tropical East Pacific Coast | Southeast Pacific Coast | Chinese Coast |
|---|---|---|---|---|---|---|---|
| $\theta_{LCL}$-$\theta_0$ in coupled PBLs (standard deviation) | 0.33K (0.36K) | 0.26K (0.39K) | 1.33K (0.74K) | 0.85K (0.86K) | 0.70K (1.34K) | 0.17K (0.28K) | 0.16K (0.93K) |
| $\theta_{LCL}$-$\theta_0$ in decoupled PBLs (standard deviation) | 8.69K (5.82K) | 2.55K (2.37K) | 3.53K (2.23K) | 2.73K (2.30K) | 10.46K (6.71K) | 1.41K (2.05K) | 3.34K (3.50K) |
| $\Delta\theta - \Gamma_m\Delta z$ above the LCL (standard deviation) | 1.22K (3.98K) | 0.07K (1.64K) | -0.39K (2.26K) | 1.65K (2.74K) | -1.06K (2.36K) | 0.48K (2.34K) | -1.16K (2.93K) |
| IS-LTS correlation | 0.53 | 0.51 | 0.35 | 0.29 | 0.43 | 0.62 | 0.62 |
| IS-EIS correlation | 0.45 | 0.58 | 0.41 | 0.36 | *-0.06* | 0.53 | 0.76 |
| IS-$EIS_p$ correlation | **0.74** | **0.76** | **0.60** | **0.48** | **0.75** | **0.74** | **0.79** |
| IS-LCC daily correlation (slope ± confidence intervals) | 0.34 (2.82 ± 0.42%/K) | 0.16 (3.05 ± 0.97%/K) | NAN | NAN | 0.26 (2.61 ± 0.43%/K) | 0.30 (2.71 ± 0.39%/K) | 0.16 (3.07 ± 0.85%/K) |
| IS-LCC monthly correlation (slope ± confidence intervals) | 0.65 (3.65 ± 0.78%/K) | 0.43 (6.44 ± 3.34%/K) | NAN | NAN | 0.38 (6.95 ± 4.02%/K) | 0.71 (4.52 ± 1.06%/K) | 0.76 (6.57 ± 1.38%/) |

As shown in section 3.2, at the ARM SGP site, the $EIS_p$ better estimates the PBL IS than both the LTS and EIS when the PBL thermal structure is largely deviated from the idealized structure of well-mixed PBLs. Next, we want to see if such a deviation exists at other radiosonde stations of subtropics and midlatitude. The ARM ENA site and other five ground-based radiosonde stations are selected to examine their characterestics of PBL thermal strucures. Their locations are shown in Fig. 7. Because the cloud base height information is not available at the radiosonde stations of IGRA, the method used at the SGP to distinguish the coupled-cloudy, decoupled-cloudy and clear sky segments is not accessible. Thus, an alternative indicator, the

decoupling degree ($\alpha_\theta$), is used to distinguish coupled and decoupled PBL according to the PBL thermal structures. The definition of $\alpha_\theta$ is introduced in Wood and Bretherton (2004) by using the liquid potential temperature ($\theta_L$) as the conserved variable during the moist adiabat. Here, $\theta$ is used to construct the moist-adiabatic conserved variable by removing the moist-adiabatic $\theta$ increase above the LCL to express the $\alpha_\theta$ parameter:

$$\alpha_\theta = \frac{\theta_{ISB}-\theta_0-\Gamma_m(z_{ISB}-z_{LCL})}{\theta_{IST}-\theta_0-\Gamma_m(z_{IST}-z_{LCL})}. \tag{12}$$

The subscripts "ISB", "IST", "0", "700hPa" and "LCL" indicate the base and top of inversion layers, the levels of 1000hPa and 700hPa and the LCL, respectively. To understand its meaning, Eq. (12) can be transformed as:

$$\alpha_\theta = \frac{\theta_{LCL}-\theta_0+[\theta_{ISB}-\theta_{LCL}-\Gamma_m(z_{ISB}-z_{LCL})]}{IS+\theta_{LCL}-\theta_0+[\theta_{ISB}-\theta_{LCL}-\Gamma_m(z_{ISB}-z_{LCL})]} \approx \frac{EIS-IS}{EIS}. \tag{13}$$

The numerator of $\alpha_\theta$ can be understood as the strength of the PBL thermal structures deviating from the coupled conditions. The denominator of $\alpha_\theta$ can be understood as the sum of the deviation strength of the PBL thermal structure from the coupled conditions and the IS (or EIS). By Eq. (13), the EIS can also be expressed as $IS/(1-\alpha_\theta)$. Thus, whether the EIS is the real IS is actually determined by the decoupling parameter $\alpha_\theta$. In perfectly coupled conditions, $\alpha_\theta$ is zero and the EIS is exactly the IS. In decoupled PBLs, when $\alpha_\theta$ is larger, the EIS actually more accounts for the deviation of the PBL thermal structure from the coupled condition. A small value of $\alpha_\theta$ would suggest a state very close to the coupled condition and here a threshold value of 0.2 is used to distinguish the coupled/decoupled PBLs based on Eq. (12). $\alpha_\theta$ has been tested for the high-resolution soundings and it comes to similar results. In fact, results listed in Table 2 at the SGP based on $\alpha_\theta$ show consistent results with that based on $\Delta z_b$.

As shown in Table 2, it is found that the two terms $\theta_{LCL}-\theta_0$ and $\Delta\theta - \Gamma_m\Delta z$ in Eq. (9) are non-negligible even over the subtropical oceans. Both the mean and standard deviation of $\theta_{LCL}-\theta_0$ are very small in the coupled PBLs. The mean of $\theta_{LCL}-\theta_0$ at the other sites in the decoupled PBLs is usually smaller (about 1-4K) as compared to that at the SGP (8.69K), except at the tropical east Pacific coast, which is larger (10.46K) than that at the SGP. Theoretically, a constant shift on the $\theta$ difference between the LCL and the ground level will not change the correlation coefficient and regression slope between the LTS/EIS and the IS/LCC. However, the term $\theta_{LCL}-\theta_0$ is systematically different between the coupled and decoupled PBLs. Thus using the LTS and EIS to sort the PBL structures will unequally mix the coupled and decoupled conditions in their different composite bins. Moreover, this bias is distinct for different places and thus the regional difference would make the LTS and EIS not uniform for their accuracies of estimating the IS. In contrast, this will not happen in the EIS$_p$ since this bias caused by the term $\theta_{LCL}-\theta_0$ in the LTS and EIS is completely excluded from the EIS$_p$.

The standard deviation of the term $\Delta\theta - \Gamma_m\Delta z$ as shown in Table 2 suggests that the errors of estimating the IS based on Eq. (9) due to the moist adiabatic assumption above the LCL of the ENA and other five radiosonde sites range from 57%-74% of that of the SGP site (3.98K). Thus the term $\Delta\theta - \Gamma_m\Delta z$ at these six sites will likely also be reduced when measuring the IS by the EIS$_p$. Thus it is not surprising that the ERA5 EIS$_p$ is best correlated with the IS directly derived from the radiosondes over all stations (Table 2). Regional differences of the correlations with the IS still exist for all metrics to measure the IS but are relatively small for the EIS$_p$.

## 4. On the relationship of global LCC with LTS, EIS and EIS$_p$

In this section, the relationship of global LCC with LTS, EIS and EIS$_p$ is discussed through daily to seasonal time scales. Since ground-based observations of radiosondes from ARM and IGRA are all assimilated in the ERA5 reanalysis (Hersbach et al., 2020), it is not surprising that the assimilated output can well capture the PBL thermal structures to estimate the IS for these locations where ground-based observations are available. However, for most areas of oceans, only limited radiosondes are available over scattered islands or during short-term campaign of field experiments to be used in ERA5 assimilation and thus whether the IS can be right captured from the ERA5 profiles needs further examination. In this section, whether the EIS$_p$

derived from the ERA5 profiles at the global scale (especially for oceans with few radiosondes assimilated into the ERA5) can better constrain LCC than LTS and EIS is explored.

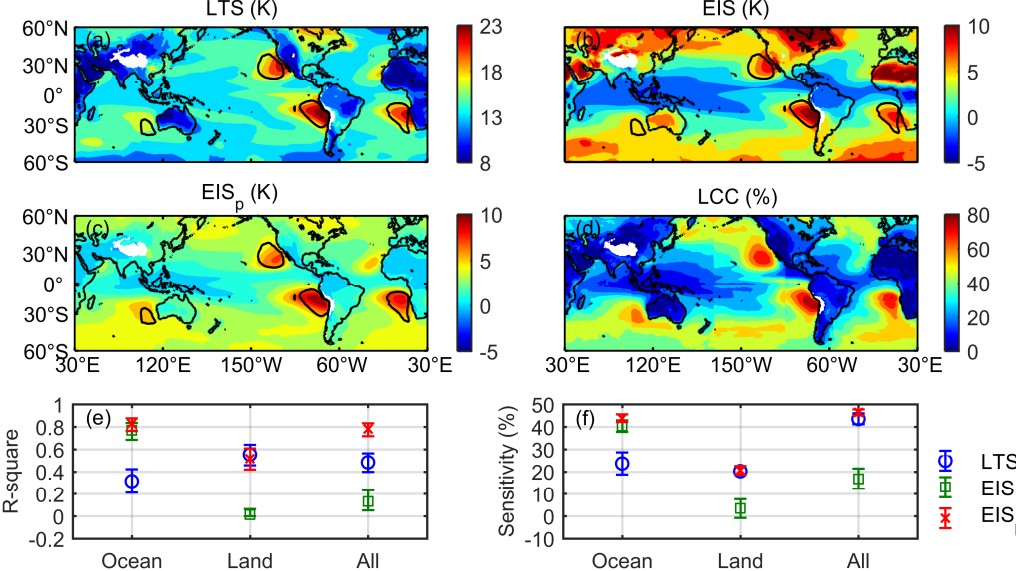

Figure 8. Spatial distribution of the ERA5 reanalysis-based LTS (a), EIS (b), EIS$_p$ (c) and the GEO-MODIS LCC (d) between 60ºS and 60ºN. The black contours enclose regions with LCC larger than 60%. The specific R-square and LCC sensitivity to the LTS (blue cycle), EIS (green square) and EIS$_p$ (red cross) over the ocean, land and all is shown in (e) and (f), respectively. The error bars show the 95% confidence interval of the mean based on the t-test.

Fig. 8 shows the six-year mean map of the ERA5-based LTS, EIS and EIS$_p$. The GEO-MODIS LCC global pattern is also used to examine its spatial correlation with the above three metrics. For the LTS, EIS and EIS$_p$, the plateau regions with the surface pressure smaller than 700hPa are not investigated here, where no GEO-MODIS LCC is observed. Overall, the annual mean value of LTS and EIS are obviously larger than the EIS$_p$ value except the inner tropical convective zone where the EIS value is large negative. In addition, there are three differences between the spatial distributions of LTS, EIS and EIS$_p$.

(1) *Over the subtropical eastern oceans*, the center locations of LTS, EIS and EIS$_p$ are different. For LTS and EIS, their center locations are more eastward and adjacent to the coast as compared with the center locations of EIS$_p$ and LCC. For EIS$_p$, its center locations are relatively away from the coast and more consistent with the center locations of LCC.

(2) *Over midaltitude oceans*, the contrast of the values between the midlatitude and the tropics is different for LTS, EIS and EIS$_p$. The midaltitude LTS reduces to the minimum but still corresponds to about 40% of LCC. The midaltitude EIS is as strong as the EIS over the subtropical eastern oceans but corresponds to the LCC much smaller than the subtropical LCC. Only the variation of EIS$_p$ from tropics to midlatitude is more reasonably consistent with the spatial variation of LCC.

(3) *Over land*, the LTS and EIS$_p$ explains over half of the LCC spatial variance according to their linear fit, but the EIS only explains 2% of the LCC spatial variance. This implies the IS is still a controlling factor for LCC distribution over land. The EIS barely correlates to continental LCC possibly because the EIS poorly estimates IS due to the strong influence of the term $\theta_{LCL} - \theta_0$ as discussed in section 3.

On the whole, the performance of EIS$_p$ is better and less dependent on surface types. Over all global oceans and land, the EIS$_p$ explains 78% of the spatial variance in LCC, significantly higher than that explained by the LTS (48%) and the EIS (13%). The spatial variations of LCC are also more sensitive to the EIS$_p$ (Fig. 8f).

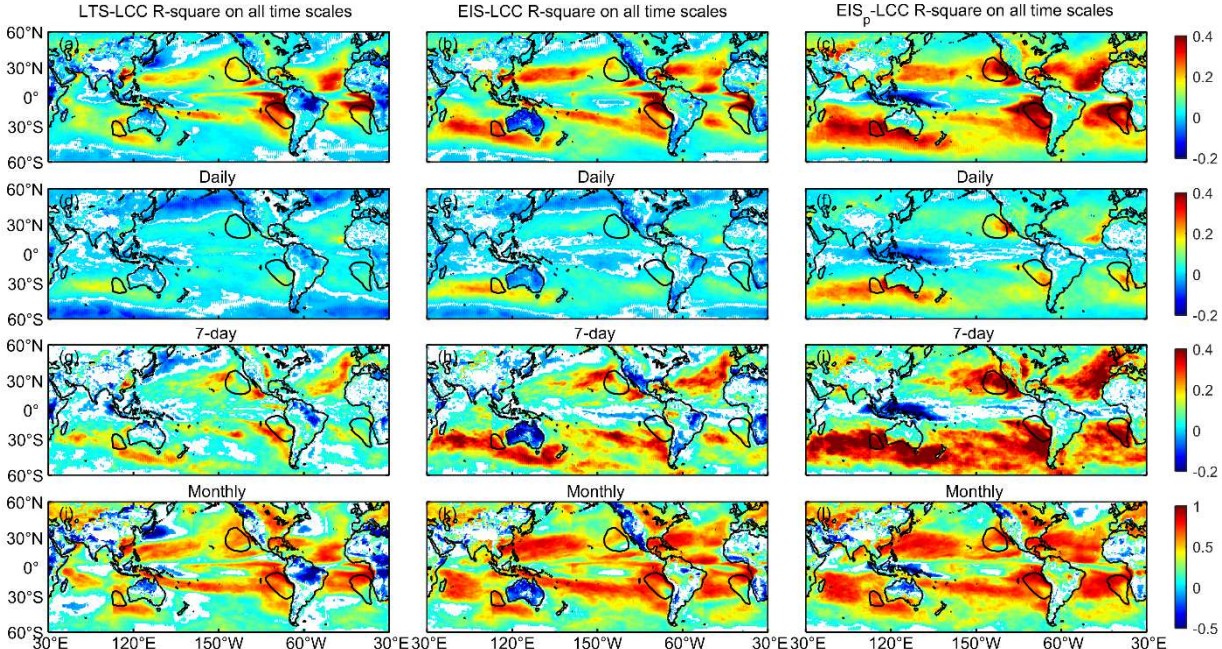

Figure 9. R-square between the GEO-MODIS LCC and the ERA5 reanalysis-based LTS (left column), EIS (middle column) and $EIS_p$ (right column) at the all-time scales (a, b and c), daily time scale (d, e and f), 7-day time scale (g, h and i) and monthly time scale (j, k and l). The black contours enclose regions with LCC larger than 60%. Only R-square at the 95% significant level are shown. The minus/plus sign of R-square indicates negative/positive correlations.

In Fig.9, the dependence of LCC on the LTS, EIS and $EIS_p$ is further examined globally for the full daily time series (i.e., all time scales) and for the daily, 7-day window-averaged anomalies and monthly means (i.e., daily, 7-day and monthly time scales). It is noted that the dependence of LCC on the three ERA5-based metrics are variant across different regions. LCC is best correlated with three metrics over the subtropical eastern oceans and some land regions that are most dominated by low clouds. Over midlatitude oceans and inner tropical convergence zone, the LCC is weakly or negatively correlated with three metrics. Thus, it is discussed separately for the most LCC-dominated regions over subtropical oceans, midlatitude oceans and land.

(1) *Over the subtropical eastern oceans with more than 60% of LCC*, on all time scales (Figs. 9a-c), the $EIS_p$ explains 36% of the variance in LCC on average, larger than that explained by the LTS (21%) and the EIS (20%). The fact that EIS does not provide a stronger correlation with LCC than LTS was also recognized by Park and Shin (2019) and Cutler et al. (2022). In contrast, the explained variance of the linear fitting between LCC and $EIS_p$ is 1.8 times of that with LTS and EIS. Besides, the mean LCC sensitivity (defined in Eq. (11) and not shown in the figure) to the $EIS_p$ on all time scales is 48% over these regions, significantly higher than that to the LTS (37%) and the EIS (36%). Although radiosondes are rare and the ERA5 profiles are mostly from the model output over these regions, the $EIS_p$ still provides a much stronger constraint on LCC than LTS and EIS. As shown in Figs. 9d-i through daily to monthly time scales, the $EIS_p$ robustly explains larger LCC variance than the LTS and EIS especially on short time scales.

(2) *Over midlatitude oceans*, weak and not significant correlations between LCC and the three metrics exist through all of time scales in Fig.9. This poor relationship is also found at the ENA site (Table 2) even using the radiosonde to derive the IS, and thus it is not caused by using the ERA5 to estimate the IS. This suggests that the IS-LCC relationship is indeed not uniform but varies with regions. Klein et al. (2017) also indicated that the LCC relationship with cloud controlling factors (e.g., the IS and sea surface temperature) is systematically different between the subtropical stratocumulus region and other regions (e.g., trade cumulus and midlatitude regions). Thus, when the IS is used to constrain the environmental influence on LCC variations, it should be noted that LCC is not all uniformly constrained by the IS for different regions. For some regions such as midlatitude oceans, the IS might not be a good constraint on LCC. But by more accurately

estimating the IS, the $EIS_p$ is more correlated with LCC than the LTS and EIS over midlatitude oceans such as North Pacific and North Atlantic on all time scales in Figs. 9a-c.

(3) *Over land regions of relatively more LCC (about 15%-25% at south America, China and Europe)*, the correlation between
540    $EIS_p$ and LCC is comparable to the subtropical oceanic regions through all of time scales in Fig.9. This suggests the $EIS_p$ is also an important controlling factor for continental LCC over these regions. Besides, the $EIS_p$ is more correlated with LCC than the LTS and EIS over most land regions, except over China where the LTS explains larger LCC variance than the EIS and $EIS_p$. The higher correlation of LTS with LCC over China might not be only attributed to the IS (LTS is not a direct measure of inversion but static stability). But more comprehensive and in-depth investigations on the LTS-LCC
545    dependence are needed to understand the exact reason of this phenomenon.

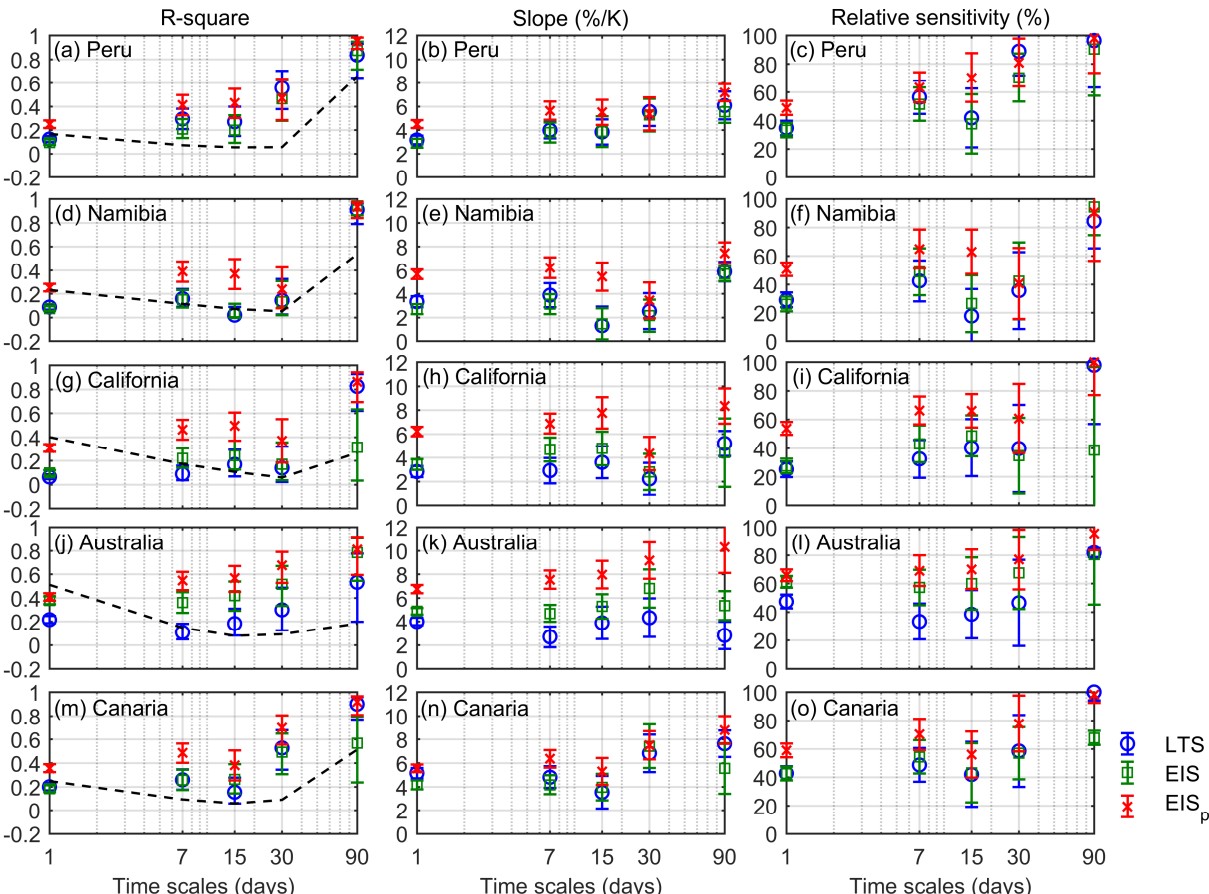

Figure 10. R-square (left panel), slope (middle panel) and relative sensitivity (right panel) of the GEO-MODIS LCC to the ERA5-based 10°×10° regional mean LTS (blue cycle), EIS (green square) and $EIS_p$ (red cross) through daily to seasonal time scales over the five typical eastern oceans defined in Klein and Hartmann (1993). The error bars show the 95% confidence
550    interval based on the t-test. The black dash lines in the left panel are the fraction of the LCC variance on different time scales divided by the total variance.

In Klein and Hartmann (1993), several key low cloud regions are defined. Those regions are of a particular interest in climate projections due to their strong low cloud albedo effects. As shown in Fig. 7, we pick eight key low cloud regions
555    according to Klein and Hartmann [1993] and the linear relationships between LCC and the three metrics are investigated. These regions lack radiosondes for long-term observations of IS. They are separated into a group of five typical tropical and subtropical low cloud prevailing eastern oceans (Fig. 10) and a group of midlatitude oceans and subtropical land (Fig.11).

As shown in Fig. 10 (the dash line in the left panel), over the five key tropical and subtropical eastern oceans, the daily and seasonal window-averaged LCC anomalies accounts for a larger portion of the total LCC variance, indicating the LCC
560    variation mainly happens at the daily and seasonal time scales. Over the Peruvian, Namibian and Canarian regions, over 50% of LCC variance are from the seasonal variations and much smaller LCC variance is from other four shorter time scales. But over the Californian and Australian regions, 40% and 51% of the LCC variance are from the daily time scale, larger than that

on other time scales. Although the LCC variance on the 7-day, 15-day and monthly time scales are relatively smaller, the sum of them still accounts for about 20~30% of the total LCC variance.

In Fig. 10, the LCC variance explained by the LTS, EIS and $EIS_p$ and the LCC slopes of the linear regression to them are examined through daily to seasonal time scales. In addition, the relative LCC sensitivity to those three metrics refers to the LCC sensitivity as defined in Eq. (11) divided by the LCC range. Here the LCC range is the difference between the mean values of the largest and the smallest 10% of LCC. The LCC variance is most explained by the $EIS_p$ among the three metrics (left panel of Fig. 10) and LCC is most sensitive to the $EIS_p$ (right panel of Fig. 10) through all of these time scales, except the monthly time scale over the Peruvian region and the seasonal time scale over the Namibian. On the daily time scale, 32% of LCC variance are explained by the $EIS_p$ on average over the five eastern oceans, which is more than twice of the variance explained by the LTS (14%) and EIS (16%). On the longer time scales (30-90 days), overall the $EIS_p$ explains 89% of the LCC seasonal variance on average over the five eastern oceans, in contrast to 80% for the LTS and 70% for the EIS. Only the $EIS_p$ can robustly explain the seasonal variance of LCC exceeding 80% for all locations. However, the EIS cannot well explain the seasonal variation of LCC over the Californian and Canarian regions, and the LTS cannot well explain the seasonal variation of LCC over the Australian region.

It is also noted that the slopes of LCC associated with each metric are not uniform across these key low cloud regions or on different time scales. A similar regional and temporal difference is also found in the LCC-IS relationships (Table 2). Klein et al. (2017) and Szoeke et al. (2016) also found the LCC slopes to the LTS/EIS is variant on different time scales and this time-scale dependence would lead to uncertainties in the final estimates of low-cloud feedbacks. Thus the error estimates of the LCC slopes to the LTS, EIS and $EIS_p$ are needed for the final uncertainty estimates of low-cloud feedbacks. To quantify the relative variation (or the uniformness) of the LCC slope to LTS, EIS and $EIS_p$, we compute the ratio between the standard deviation and the mean of grouped slopes. For the temporal relative variation, over each region slopes on different time scales are grouped together. While for the regional relative variation, on each time scale slopes over different regions are grouped together. The temporal relative variation of the LCC slope to the LTS and EIS is 32% and 29% on average over the five eastern oceans. In contrast, the temporal relative variation of the LCC slope to the $EIS_p$ is 21%. Besides, the regional relative variation of the LCC slope to the LTS, EIS and $EIS_p$ is 24%, 21% and 18% between the five eastern oceans, respectively. This suggests that the regional/temporal dependence of the LCC slope in the estimate of low cloud feedbacks is also non-negligible and needs to be considered in the final error estimates or to estimate low-cloud feedbacks by separating regions.

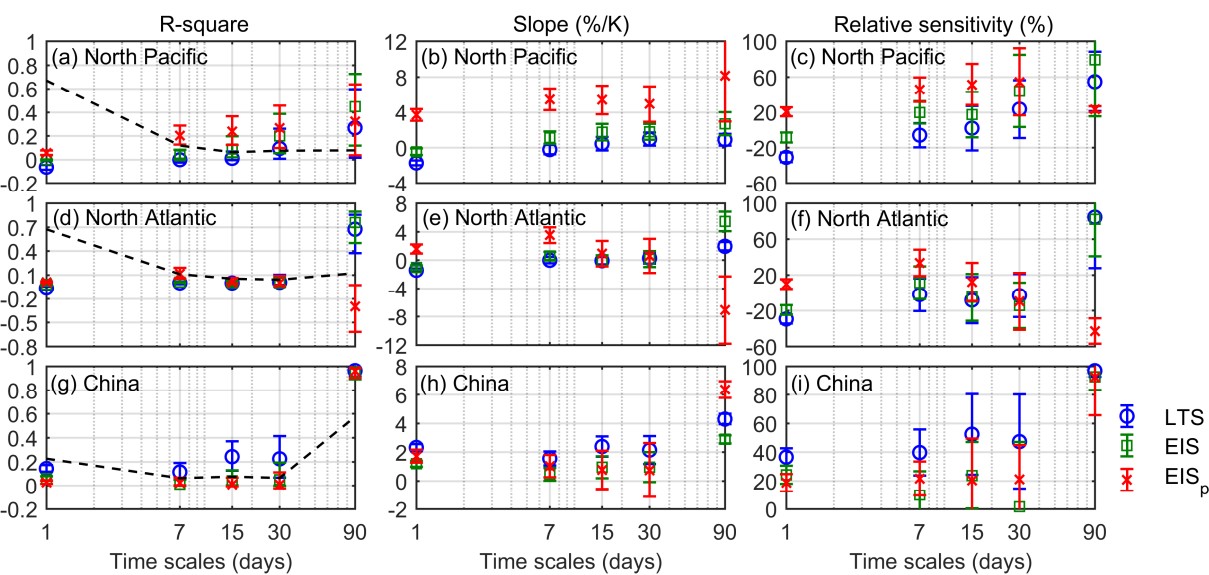

Figure 11. Similar to Fig. 10 but for the other three regions defined in Klein and Hartmann (1993), including two midlatitude oceans and one subtropical land. The minus/plus sign of R-square indicates negative/positive correlations.

Figs. 11a and 11d (the dash line) show that over the North Pacific and North Atlantic regions, 67% of the LCC variance is from the daily time scale, while over the China region in Fig.11g, variance is mostly from the seasonal time scale (57%). Over the North Pacific and North Atlantic regions, LCC is not necessarily correlated with the IS. Norris (1998) has found that fogs and bad-weather stratus clouds frequently occur over the midlatitude ocean but with less inversion and poor IS-LCC relationships. Similarly, poor correlations (Figs. 11a and 11d) and sensitivity (Figs. 11c and 11f) between LCC and LTS/EIS/$EIS_p$ are found over the North Pacific and North Atlantic. But, the $EIS_p$ is closest to the radiosonde-detected IS as compared with the LTS and EIS at the ENA and OWS C as shown in Table 2. This suggests that the $EIS_p$ is still a reliable estimation for the IS to represent the true IS-LCC relationship. But the LTS/EIS-LCC relationship is not necessarily due to the IS influence on LCC. Figs. 11a, 11d, 11c and 11f also show that the LTS/EIS-LCC correlation and sensitivity is very different from that between LCC and $EIS_p$ on the daily and seasonal time scales. Unfortunately, it has not been well explored about the midlatitude LCC-IS relationship. The poor $EIS_p$-LCC relationship represents that the IS cannot be a cloud controlling factor as important as that over subtropical oceans. Over the Chinese region, the EIS and $EIS_p$ are both better correlated with the IS as shown in Table 2. Fig. 11g and Fig. 11i shows that LCC is slightly more corelated with and sensitive to the LTS through all time scales. These higher correlations and sensitivity are not related to the IS since the LTS correlates the least with the IS. Since the LTS not just includes the IS but actually represents the total static stability from 1000hPa to 700hPa to influence the amount and liquid water path of low clouds (Klein and Hartmann, 1993; Kawai and Teixeira, 2010), it may imply that there are other thermal factors in addition to the IS in LTS to contribute to these higher correlations and sensitivity. Overall, it should be noted that the IS may not be a strong cloud-controlling factor over the midlatitude oceans and subtropical land but $EIS_p$ is still the best estimation for the IS. The IS is not the only LCC controlling factor, and other factors (e.g., sea surface temperature, cold advection, free-tropospheric humidity and vertical velocity) are also important for influencing LCC (Myers and Norris, 2013; Klein et al., 2017).

All above analyses (Figs. 8-11) are based on the daily-averaged LTS, EIS and $EIS_p$ data, which are computed based on the 3-hour 1° ERA5 atmospheric profiles. Based on the monthly mean atmospheric profiles, over the region of LCC larger than 60%, the LTS and EIS explain 50% and 48% of LCC variance, which is similar to the value of 53% based on the 3-hour ERA5 atmospheric profiles. However, the $EIS_p$ based on the monthly mean ERA5 profiles explains 49% of the LCC variance, which is significantly lower than the 65% based on the 3-hour profiles. Thus for accurately computing the $EIS_p$ on either short or long timescales, high temporal resolution of reanalysis data is necessary.

## 5. Conclusion

In this paper, a novel profile-based estimated IS ($EIS_p$) is developed based on the thinnest possible layer that contains the inversion layer in the ERA5 profiles. By this method, the effects of the static stability below the LCL are completely removed. The errors due to the spread of the environmental $\theta$ gradient around the moist adiabat above the LCL are reduced.

At the ARM SGP site, the $EIS_p$ more accurately estimates the IS, with a correlation of 0.74, than the LTS (0.53) and EIS (0.45). Thus, the $EIS_p$ reasonably replicates the constraints of IS on the PBL moisture distribution and LCC, while the LTS/EIS has a weak/erroneous relationship with the PBL moisture and LCC. The LCC sensitivity to LTS and EIS and $EIS_p$ is 39%, 12% and 50%, respectively. On the daily time scale (7-day mean excluded), the variance in LCC explained by the $EIS_p$ (9.1%) is more than twice that explained by both the LTS (3.1%) and EIS (-0.4%). At the ARM ENA site, the $EIS_p$ has similar advantages on estimating the IS. At other available oceanic and coastal observation stations, the $EIS_p$ is still a better estimation for the IS than the LTS and EIS.

At the global scale, according to the GEO-MODIS LCC observations, the $EIS_p$ better explains the spatial and temporal variations of LCC than the LTS and EIS. Over oceans, the $EIS_p$ distribution is more consistent with the LCC pattern compared with the LTS and EIS. The locations of the strongest $EIS_p$ are consistent with the centers of the largest LCC relatively away

from the coast, while the centers of the strongest LTS and EIS are over the coast. Over the subtropical LCC domains, the LCC sensitivity to the $EIS_p$ is 48%, larger than that to the LTS (37%) and EIS (36%) on all time scales. And the increased LCC sensitivity to $EIS_p$ primarily comes from time scales shorter than a month. Over the typical low-cloud prevailing eastern oceans as defined in Klein and Hartmann (1993), the LCC daily variance explained by the $EIS_p$ is 32% and twice that explained by the LTS/EIS. And the LCC seasonal variance explained by the $EIS_p$ increases to 89% as compared with that explained by the LTS (80%) and EIS (70%).

No uniform relationship between the LCC and any of the IS, LTS, EIS and $EIS_p$ is found across time scales or different regions. As compared to the LTS/EIS, the temporal relative variation of the LCC slopes to the $EIS_p$ is reduced from 32%/29% to 21%. The regional relative variation of the LCC slope to the $EIS_p$ is slightly smaller than that of LTS and EIS. This non-uniform LCC sensitivity to cloud controlling factors across different regions and time scales suggests that using a single observational multi-linear regression between LCC and cloud-controlling factors to estimate the global low cloud feedbacks is not recommended.

Overall, the $EIS_p$ is an improved measure of the IS and better constrains LCC, especially on time scales shorter than a month. On short time scales, the enhanced dependence of LCC on the $EIS_p$ makes the $EIS_p$ more suitable to resolve process-oriented studies associated with LCC variations. Therefore, the $EIS_p$ is likely a better constraint to reduce the meteorological covariations to separate the aerosol effects in aerosol-cloud interactions.

**Author contribution.**

JY and RW designed the experiments and ZW carried them out. JY and ZW prepared the first version of the manuscript with contributions from all co-authors. YC prepared the ERA5 data and TT inspected some individual profiles and cloud images. All authors verified the final version of the manuscript.

**Competing interests.**

The authors declare that they have no conflict of interest.

**Acknowledgment**

This work was supported by the NSFC-41875004 and the National Key R&D Program of China (2016YFC0202000). The first author thanks the "Double First-class" initiative program providing an opportunity for him to visit and study at the University of Washington.

**Data Availability Statement**

All data used in this study are available online. The ARM SGP and ENA radiosonde and cloud observations were obtained from the ARM Research Facility and are available at https://www.arm.gov. The IGRA radiosondes are available from the NOAA National Centers for Environmental Information at https://www.ncei.noaa.gov/products/weather-balloon/integrated-global-radiosonde-archive. The GEO-MODIS LCC product is provided by the NASA Langley Research Center at https://earthdata.nasa.gov. The ERA5 reanalysis used in this study is from the ECMWF and available at https://cds.climate.copernicus.eu/cdsapp#!/home.

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
