# Peer review of "Profile-based Estimated Inversion Strength"

_Atmospheric Chemistry and Physics, 2022_

## Referee Comment (RC1)

The manuscript is focused on approximations of the inversion strength in situations when low level clouds occur. These approximations are designed to estimate the real inversion strength using relatively coarse spatial and temporal resolution datasets such as a reanalysis. Two metrics in particular have been proposed as a means to relate cloud cover to the prevailing meteorological conditions, that relate to inversion strength, one is the Lower Tropospheric Stability parameter (LTS; Klein and Hartmann, 1993) and the other the Estimated Inversion Strength (EIS; Wood and Bretherton, 2006). LTS is the difference in potential temperature between 700 hPa and the surface while EIS is a modification of LTS where the potential temperature change along the moist adiabat above the lifting condensation level is removed. The authors discuss situations where neither LTS nor EIS are a good fit for the real inversion strength and propose a correction to EIS where the variations in potential temperature below the LCL are removed to avoid a strong deviation from the real inversion strength in situations when the PBL is decoupled, as well as above the inversion to avoid errors caused by deviations from the moist adiabat over a relatively wide distance above the LCL. Using high resolution radio soundings at a site in continental United States, they demonstrate how their estimated inversion strength 1) performs better than EIS or LTS for characterizing the actual inversion strength and 2) offers a stronger correlation with low level cloud cover. They then explore how the three metrics relate to clouds geographically as well as for different temporal resolutions. The whole analysis is extensive, very thorough and captivating, but covers a lot of grounds and can be overwhelming at times. However the demonstration and justification for this new parameter is convincing and will be a useful addition in the quest for the most important cloud controlling factors. The figures are well done and compelling while the verification process precise and careful, so in essence, this is a good manuscript and there is no need for additional scientific work. But to become really accessible, the presentation of the results could be much improved and should be simplified with some reorganization.

Therefore before the manuscript be accepted for publication, there are a number of improvements to be made. My main comments and suggestions are as follows, with more specific comments listed below:

- While the work is logically presented, there are some presentation issues. There is quite a lot of ground covered by the paper, maybe this could be streamlined. There are two aspects in this analysis that are covered: 1) which metric can approximate inversion strength the best and 2) where and at which temporal resolution does inversion strength correlate with low level cloud cover. The separation between the two should be clearer to help follow the narrative, and avoid the repetition of how much of an improvement the EISp metric is.
- 2) The definition and construction of the new inversion strength estimate proposed here is very confusing, I strongly encourage the authors to rethink their explanations in section 3. In fact this crucial section could be much improved if it is divided into a first part that explains the theory behind EISp and how it is calculated. Then a second part that explores how it relates to inversion strength, and a comparison with LTS/EIS respective relations to IS using the SGP data. Finally this can then be expanded with a comparison to IS using the IGRA and ENA data. Once you have made the case that EISp is a superior measure of inversion strength, you can move on to a section 4 that explores its relationship with clouds.

- 3) There is little discussion of the expansive work and knowledge accumulated on the physical reasons for a strong link between LTS or EIS and cloud cover, so sometimes it seems that the fact that this link exists comes as a surprise. I made a few suggestions below of papers that would help in the interpretation of the results, some are already cited.
- 4) The distinction and classification of coupled, decoupled and clear sky profiles is confusing, there are different metrics used and these appear at different points in the manuscript, which is quite confusing. Because this distinction is at the center of the whole argument in favor of EISp, the matter merits its own section. A subsection in section 2 dedicated to the subject would be a helpful addition and thereafter the method would not need to be explained again.
- 5) The language can be confusing at times, and if simplified this should help a great deal follow the argument. Some suggestions of changes are given in both lists below.

**Specific comments:**

- Line 9: "At the southern great plains site" needs some additional information such as where it is, and which agency/project runs it. Given this is the abstract, it might be easier to replace here with something similar to "at a ground-based site in north America" for example.
- 2. Line 53-54: if inversion strength is not well predicted by LTS or EIS at high temporal resolution but quite well at longer time scales, what could be the reason? This should be discussed.
- 3. Line 56-57: this sentence is unclear, especially "may not be conserved but with a stable layer". Please rephrase.
- 4. Line 70: "ARM" has not been introduced yet, what is it and which agency runs this? For the title of the subsection I suggest replacing "ARM sites" with "ground-based sites" or a short introduction of section 2 detailing the sort of data to be presented would be great and it could be used to introduce ARM etc..
- 5. Line 71: so here "SGP" is used, I do not think it has ever been said it means Southern Great plains. Also why use this site? It is inland and therefore quite different from the environments usually associated with low clouds such as the subtropical stratocumulus regions. Why isn't ENA your primary site instead? This needs to be explained at the outset.
- 6. Line 72: "the ARM" is odd, just "ARM" would suffice. But it should really be "established by the Atmospheric Radiation Measurement program of the Department of Energy", unless it is already said in an intro (see comments above).
- 7. Line 95: do you keep profiles that have low clouds but with other clouds aloft also present?
- 8. Line 97-99: here you mention separating coupled and decoupled PBL conditions but do not explain how it is done until much later in the paper. So either explain the method here or refrain from introducing the classification until it becomes relevant.
- 9. Line 102-103: the sentence "our results..." is unclear and incorrect. How about replacing with "this will be verified and discussed later".

- 10. Lines 103-107: finally the reason for choosing SGP over ENA is explained but the explanation is confusing. And sounds very ad hoc and not very scientific. I think that your point is that the work focuses on the strong relationship between cloud cover and inversion strength, but this relationship is not true everywhere and in particular it is not observed at the ENA or weak. But it is found at the SGP, making this location a good place to explore a refinement of the inversion strength estimate. I wonder why the ENA site is mentioned at all?
- 11. Line 115: another acronym that is not defined: "OWS"
- 12. Line 170: "as the height of three fourth of the greatest ...." Needs to be rewritten, it is really hard to understand what is done.
- 13. Line 172: "these two methods both works" is incorrect, consider replacing with "these two methods work" or "these methods both work", BUT either way, it is confusing: does this mean the two methods give the same result? Which method do you actually use? Why mention two if you only use the one?
- 14. Line 176-179: the technique used to get the inversion strength for coarse vertical resolution radiosoundings is confusing, please clarify.
- 15. Line 188: replace "in consistent with that" with "as"
- 16. Line 193: as for section 2, a short intro in section 3 would help situate where the analysis is going next and why.
- 17. Line 224: why use "180 m" and not "150m" as in Jones et al? not clearly explained.
- 18. Line 225: so is this method the same as that used in Jones et al 2011 or something else? If it is then just say so. If it is not the same, then justify it.
- 19. Line 228: "large positive skewness": not sure I understand why this is? This should be discussed.
- 20. Line 230-231: this sentence is not entirely clear nor correct. By construction, LTS and EIS will include the term (theta\_LCL-theta\_0) regardless of its value. What you're trying to say I think is that in situations where this term is not zero, such as clear or decoupled PBLs, this will cause LTS and EIS to deviate from the real inversion strength value. No? Please rephrase accordingly.
- 21. Line 232: "even in the coupled PBLs: do you mean "decoupled" instead? Or is "even" misplaced? Unclear.
- 22. Line 241: "could easily overwhelm the real IS" is weird. What you simply mean is that this term will cause an overestimate of the IS estimated with LTS or EIS, so why not rephrase this sentence with this in mind.
- 23. Line 259-260: this sentence seems very obvious, no clouds present suggests adverse conditions for cloud formation indeed. Not sure what the point is?
- 24. Figure 2 and discussion: this is for all seasons for the full period with SGP data, correct? Might want to remind the reader.
- 25. Line 282-283: I wonder if the better relationship between RH and LTS compared to EIS does not come from the fact that EIS is meant to approximate inversion strength so necessitates that an inversion be present, when LTS is meant to represent static stability and can still relate to clouds when there is no inversion (or very weak). Have you tried to check LCC vs EIS by separating situations with IS >0 from those with IS=0 or <0?

- 26. Line 339: for the ERA5 vs SGP sonde comparison, what is the time resolution of the ERA5? 3-hourly or hourly? Is the same time of day used?
- 27. Line 343: "while decreasing RH..." is unclear, please clarify.
- 28. Line 346: language not very scientific, replace "does a better job on estimating the IS" with "offers a better fit to the real IS"
- 29. Line 373: language not very rigorous, replace "not bad" with e.g. "remarkably close to that explained by IS"
- 30. Section 4: again an intro would help follow the narrative.
- 31. Section 4.1, title not exact, only a few locations are chosen, this is not true worldwide, please rephrase.
- 32. Table 2, caption: what does "excluding 7-day means" mean?
- 33. Line 390: "globally" is again too vast, maybe "in other locations" or "in maritime locations" would be more appropriate?
- 34. Line 391: here the coupled vs decoupled distinction is done again, and cloud base height is mentioned (as having been used earlier). This was not clear. Maybe best to always use the same definition no? could this alpha parameter be tested for the high resolution soundings as well?
- 35. Line 396: unclear, is this parameter introduced by Wood and Bretherton? Or by you but happens to be consistent to what they had done?
- 36. Line 397: "the decoupling degree is in proportion..." is awkward. You simply mean that when alpha is zero the PBL is coupled, and when it is not, it is decoupled. Because a very small value suggest a state very close to coupled, you use 0.2 as a threshold to separate the two conditions. Please rewrite this paragraph in a similar fashion if my interpretation is correct. That being said, a physical explanation of what the formula (#12) is expressing is missing and should be added to help understand what it represents.
- 37. Line 424: I am not sure why the second sentence only mentions EISp, clearly there are no cloud data in high elevations regions, namely Tibetan Plateau and Andes (maybe others but these two are the most obvious) so just say so, the reader can see the analysis cannot be done in these areas for sure.
- 38. Line 425-432: this whole discussion of Fig 8 is unclear and hand-wavy at best. I cannot clearly see what the authors are trying to convey here. It might help to overlap the LCC contours (as solid lines) on the LTS/EIS/EISp maps and possibly draw boxes around the areas under focus here. But the whole paragraph needs clarification for sure.
- 39. Line 436: "poor fault tolerance" is unclear, what does this mean?
- 40. Line 438: language not very rigorous, consider replacing "but these will not happen with the EISp" with "The performance of EISp is less dependent on surface type" (assuming that this is what is meant).
- 41. Line 438-439: the whole sentence could equally say that LTS can explain LCC variations over land and EIS over ocean nearly as well as EISp. While it is clear EISp is performing better and more uniformly, your analysis still demonstrates where LTS and EIS have advantages and explains why they have been relied upon for quite some time.
- 42. Line 449/Fig 9: what is "all time scales" exactly??

- 43. Line 459: you could add reference to the Klein et al. 2017 review that summarizes all the work done on the environmental factors that impact LCC in addition to IS. And this diversity explains the non geographical uniformity of the effect these parameters have on LCC, and in particular explains your result that "IS-LCC relationship is not uniform but varies with regions even over oceans". I would add that SST plays an important role and it is not uniform either.
- 44. Line 468: I would suggest revisiting the work of Klein and Hartmann (1993) to better understand why LTS and LCC are related despite LTS not being well correlated with IS (again LTS is not a direct measure of inversion but of static stability).
- 45. Line 490: what is sensitivity here, the slope?
- 46. Lines 496-498: I do not understand the point of this last sentence, just that EISp is the only metric that exceeds 80% of the variance explained for all locations? Please consider simplifying the sentence.
- 47. Line 525: again the sentence "Unfortunately.." is unclear, what is this discussing exactly? is little known about this because it has not been explored or the authors do not have a satisfying explanation? "Limited knowledge" implies there is some, so what is it?
- 48. Lines 529-530: it might be useful to remind the reader what the actual definition of LTS is and remind what KH93 had in mind. Also see Kawai and Teixeira (2010, JCLI, doi: 10.1175/2009JCLI3070.1).
- 49. Line 530-531: again, see comment above, IS is not the only cloud controlling factor, see also Myers and Norris (2013, JCLI, doi:10.1175/JCLI-D-12-00736.1), cited in your manuscript.
- 50. Line 564: "questionable" should be replaced with "not recommended", but as conveyed in the comments above, there is a large amount of literature on the role of various meteorological metrics on cloud properties, I doubt that cloud feedback is estimated with only inversion strength in mind. But if it is, a citation should be added.

**Typos/grammar:**

- 1. Line 101: remove "oceans", redundant.
- 2. Line 144: "away about 2.8km" is incorrect, replace with "within about 2.8 km"
- 3. Line 148: here and in other places, "metrices" should be "metrics"
- 4. Line 173: remove "s" at end of "jump" and "increase"
- 5. Line 208: "could compute" is odd, do you mean "would include"?
- 6. Line 209: "as the IS estimates" is awkward, do you mean "in the IS estimates"?
- 7. Line 209: Remove "in general", awkward.
- 8. Line 229: "strength" is not great, how about "value"?
- 9. Line 297: remove "that" at the beginning of this line.
- 10. Line 320: replace "three" after "among" with "them".
- 11. Line 341: "close" should be "closer"
- 12. Line 392: insert "parameter" at the end of the sentence (before the ":")
- 13. Line 395&398: I believe "2004" should be "2006", unless there is a missing reference in the list.
- 14. Line 488: "composites LCC to them" is odd, maybe "the slopes of the linear regression with LCC"?

- 15. Line 489: "besides" should be "in addition" I think
- 16. Line 499: "cross" should be "across"
- 17. Line 529: "the worst estimation of the IS" should be "correlates the least with IS"
- 18. Line 545: "wrong" should be "erroneous"

---

## Author Comment (AC1)

**Response to Anonymous Referee #1**

*Reviewer #1: The manuscript is focused on approximations of the inversion strength in situations when low level clouds occur. These approximations are designed to estimate the real inversion strength using relatively coarse spatial and temporal resolution datasets such as a reanalysis. Two metrics in particular have been proposed as a means to relate cloud cover to the prevailing meteorological conditions, that relate to inversion strength, one is the Lower Tropospheric Stability parameter (LTS; Klein and Hartmann, 1993) and the other the Estimated Inversion Strength (EIS; Wood and Bretherton, 2006). LTS is the difference in potential temperature between 700 hPa and the surface while EIS is a modification of LTS where the potential temperature change along the moist adiabat above the lifting condensation level is removed. The authors discuss situations where neither LTS nor EIS are a good fit for the real inversion strength and propose a correction to EIS where the variations in potential temperature below the LCL are removed to avoid a strong deviation from the real inversion strength in situations when the PBL is decoupled, as well as above the inversion to avoid errors caused by deviations from the moist adiabat over a relatively wide distance above the LCL. Using high resolution radio soundings at a site in continental United States, they demonstrate how their estimated inversion strength 1) performs better than EIS or LTS for characterizing the actual inversion strength and 2) offers a stronger correlation with low level cloud cover. They then explore how the three metrics relate to clouds geographically as well as for different temporal resolutions. The whole analysis is extensive, very thorough and captivating, but covers a lot of grounds and can be overwhelming at times. However the demonstration and justification for this new parameter is convincing and will be a useful addition in the quest for the most important cloud controlling factors. The figures are well done and compelling while the verification process precise and careful, so in essence, this is a good manuscript and there is no need for additional scientific work. But to become really accessible, the presentation of the results could be much improved and should be simplified with some reorganization.*

**Response:** We greatly appreciate the reviewer's comprehensive and very valuable comments and suggestions. That helps the authors significantly improve the representation of this manuscript. We have carefully taken these comments into account and accordingly revised and reorganized the manuscript to make it more accessible.

*Therefore before the manuscript be accepted for publication, there are a number of improvements to be made. My main comments and suggestions are as follows, with more specific comments listed below:*

1) *While the work is logically presented, there are some presentation issues. There is quite a lot of ground covered by the paper, maybe this could be streamlined. There are two aspects in this analysis that are covered: 1) which metric can approximate inversion strength the best and 2) where and at which temporal resolution does inversion strength correlate with low level cloud cover. The separation between the two should be clearer to help follow the narrative, and avoid the repetition of how much of an improvement the EISp metric is.*

**Response:** Thanks for suggestions to help us reorganize the manuscript and improve the representation. The presentation of results has been reorganized as follows:

a) *Section 3. The profile-based method of EIS (EISp)*. This section introduces the new EISp

and demonstrates that the EISp can approximate inversion strength the best.

**b)** *Section 4. On the relationship of global LCC with LTS, EIS and EISp*. This section presents the correlation between LCC and LTS, EIS and EISp and demonstrates the EISp is a better cloud-controlling factor.

Brief introductions have been added at the beginning of section 3 and section 4. The main focus of each section is then introduced to help follow the narrative (please see responses to specific comments 16 and 30).

The analyses of the results in section 4 have been separated into three parts to discuss according to regions: (1) subtropical eastern oceans; (2) midlatitude oceans; (3) Land.

*2) The definition and construction of the new inversion strength estimate proposed here is very confusing, I strongly encourage the authors to rethink their explanations in section 3. In fact this crucial section could be much improved if it is divided into a first part that explains the theory behind EISp and how it is calculated. Then a second part that explores how it relates to inversion strength, and a comparison with LTS/EIS respective relations to IS using the SGP data. Finally this can then be expanded with a comparison to IS using the IGRA and ENA data. Once you have made the case that EISp is a superior measure of inversion strength, you can move on to a section 4 that explores its relationship with clouds.*

**Response:** As suggested, the section 3 has been reorganized into three subsections accordingly as follows:

**a)** *Section 3.1 The algorithm of the new EISp*. In this subsection, the new profile-based method of EIS is introduced;

**b)** *Section 3.2 PBL stratification and the establishment of the EISp at the SGP*. In this subsection, the PBL thermal structure is examined and how the deviation of PBL thermal structure from the well-mixed condition influences IS estimates is discussed, which provides the necessity of the new EISp and observational evidences about why the EISp is a better estimate for the IS. The accuracy of the EISp representing the IS and the efficiency of constraining LCC are examined by comparing it against the radiosonde-measured IS.

**c)** *Section 3.3 Validation of EISp at radiosonde stations of subtropics and midlatitude*. In this subsection, the $EIS_p$ is further validated at several radiosonde stations of subtropics and midlatitude.

A brief introduction has been added at the beginning of section 3 to state the focus of each subsection (please see the response to the specific comment 16).

The definition and construction of the new EISp has been rewritten in section 3.1 as:

**Line 226-255 in the revised manuscript,** "The EISp is designed to capture the IS information from the thinnest layer encompassing the inversion in low-resolution (hundreds of meters) atmospheric profiles. For these coarse-resolution profiles (e.g., ERA5), it is difficult to accurately locate the exact place of the inversion because usually the thickness of the inversion is much smaller than the distance between two adjacent vertical levels. Thus only one or two adjacent layers that could encompass the inversion are located. The latter is for

the consideration that an inversion layer may be across two adjacent layers of the ERA5. Specifically, the EISp is computed as follows:

(1) Locating the layer of the maximum $\theta$ vertical gradient $(d\theta/dz)_{max}$:

For each hourly ERA5 profile, the layer of $(d\theta/dz)_{max}$ is firstly located between the LCL and 5km AGL (the red zone in Fig.1), since the inversion just features strong gradients in thermodynamical properties.

(2) Finding the layers encompassing the full inversion:

The layer of $(d\theta/dz)_{max}$ may not encompass the full inversion if the inversion crosses two adjacent layers of the ERA5. Thus, the layer of $(d\theta/dz)_{max}$ is combined with an adjacent layer just above or below it respectively, to constitute other two candidate layers that could encompass the full inversion (the blue and green zone in Fig.1).

(3) Calculating the EISp:

The EISp is calculated for the three possible layers identified in second stage, respectively:

$$EIS_p = \theta_{top} - \theta_{base} - \Gamma_m(z_{top} - z_{base}), (8)$$

where subscripts "top" and "base" represent the top and base levels of a candidate layer. $\Gamma_m$ is computed using Eq. (5) at the base level. The $\theta$ increase of the moist adiabat is removed to extract the strength of the inversion between the top and base levels, which is consistent with the EIS framework in Wood and Bretherton (2006). The final EISp is determined by which layer in Fig.1 encompasses stronger inversion computed from Eq. (8) and thus refers to the largest value among the three candidates EISp1-3.

The EIS (Wood and Bretherton, 2006) assumes that the PBL is well mixed (dry adiabat below the LCL and moist adiabat above the LCL) for estimating the IS. If that is the case, EISp would give the same results as EIS. However, it will be shown in the following sections that the actual PBL often deviates from the well mixed conditions, where the EISp provides a physically more reasonable estimate for the IS than the EIS and thus a stronger cloud-controlling factor.

When high-resolution radiosondes are available, the exact IS can be obtained fairly straightforward (section 2.5, Eq. 6). The computation of EISp is in fact adapted from the algorithm of obtaining the IS from high-resolution radiosondes, but is adjusted to suit coarse-resolution atmospheric profiles in reanalysis. Because high-resolution soundings are rare, an applicable metric derived from reanalysis would be much more beneficial. Because the IGRA soundings have similar vertical resolutions as ERA5 in lower troposphere, the IS of these soundings (used in section 3.3) is derived exactly by the same way as the EISp.".

*3) There is little discussion of the expansive work and knowledge accumulated on the physical reasons for a strong link between LTS or EIS and cloud cover, so sometimes it seems that the fact that this link exists comes as a surprise. I made a few suggestions below of papers that would help in the interpretation of the results, some are already cited.*

**Response:** The physical explanation for the strong link between LTS or EIS and cloud cover has been added in the introduction:

**Line 25-28 in the revised manuscript,** "Strong IS inhibits the dry air above the inversion from being incorporated into the PBL and traps moisture below the inversion to favor greater cloud cover (Wood and Bretherton, 2006; Mauger and Norris, 2010). In contrast, weak IS promotes the drying effect of entrained air from the free troposphere and reduces the PBL

moisture to decrease cloud cover (Bretherton and Wyant, 1997; Myers and Norris, 2013).".

*4 ) The distinction and classification of coupled, decoupled and clear sky profiles is confusing, there are different metrics used and these appear at different points in the manuscript, which is quite confusing. Because this distinction is at the center of the whole argument in favor of EISp, the matter merits its own section. A subsection in section 2 dedicated to the subject would be a helpful addition and thereafter the method would not need to be explained again.*

**Response:** The description about how to distinguish coupled cloudy, decoupled cloudy and clear sky profiles has been moved to section 2.1:

**Line 117-127 in the revised manuscript**, "These hourly segments are further sorted into three categories: clear sky, coupled cloudy and decoupled cloudy segments. Clear sky segments are those in which no cloud is present within that segment. The coupled/decoupled cloudy segments are segments containing low-clouds in coupled/decoupled PBLs, respectively. A straightforward indicator to distinguish coupled and decoupled PBLs is the height difference between the cloud base and the LCL ($\Delta z_b$) (Jones et al., 2011). When the PBL is well mixed, $\Delta z_b$ is close to zero, but in the decoupled PBLs the cloud and subcloud layers would be separated by a stable layer and the LCL may diverge from the cloud base hundreds of meters with large $\Delta z_b$ (Nicholls, 1984; Jones et al., 2011). The threshold value of $\Delta z_b$ is empirical and for different instrument capability, vertical resolution and locations the threshold may be a little different. In reference to the linear least-square fit between $\Delta z_b$ and $\Delta\theta$ in Jones et al. (2011) that 150 meters of $\Delta z_b$ correspond to 0.5K of the $\theta$ difference in the subcloud layer, a similar linear relationship is found but the slope is a little different that 180 meters of $\Delta z_b$ corresponds to 0.5K of the $\theta$ difference at the SGP site. Thus at the SGP site, a threshold value of 180 meters for $\Delta z_b$ is used to distinguish coupled and decoupled PBLs.".

The clear-sky segments include both coupled and decoupled PBLs and this has been clarified:

**Line 285-286 in the revised manuscript,** "Note that the $\Delta z_b$ method cannot distinguish whether the PBL is coupled or decoupled when a segment has no low cloud. Thus the clear sky segments might contain both coupled and decoupled PBL".

*5 ) The language can be confusing at times, and if simplified this should help a great deal follow the argument. Some suggestions of changes are given in both lists below.*

**Response:** The confusing sentences have been rewritten or simplified according to the reviewer's comments. Thanks for helping us to improve the representation.

***Specific comments:***

*1 . Line 9: "At the southern great plains site" needs some additional information such as where it is, and which agency/project runs it. Given this is the abstract, it might be easier to replace here with something similar to "at a ground-based site in north America" for example.*

 **Response:** It has been changed as:

**Line 9-10 in the revised manuscript,** "at a ground-based site in north America".

*2.   Line 53-54: if inversion strength is not well predicted by LTS or EIS at high temporal resolution but quite well at longer time scales, what could be the reason? This should be discussed.*

**Response:** The accuracy of using LTS or EIS to predict inversion strength relies on whether the PBL thermal structure resembles the $\theta$ profile of the well-mixed condition (dry adiabat below the LCL and moist adiabat in the free troposphere). With a larger spread of the bias at high temporal resolution, larger uncertainty would be added to LTS and EIS. On the monthly time scales, the bias of the averaged $\theta$ profiles deviating from the dry/moist adiabat below/above the LCL has less monthly variation, which imposes less influence on predicting the monthly variation of inversion. The reason that on shorter time scales the deviation of PBL thermal structure from the well-mixed condition is likely due to the turbulent and chaotic nature of PBL. As shown in Fig. 2 in the revised manuscript (original Fig. 1), if averaged over longer periods, the spread of temperature stratification of PBL is likely reduced.

However, this can explain lower correlations (more noise in the data) between three metrics and IS or LCC, but cannot explain the differences in regressed slopes (sensitivity of 3 three metrics to LCC) across different time scales. The time-scale dependence of LCC sensitivity to LTS/EIS has also been recognized by Szoeke et al., (2016). But up to now, the exact reason why the LCC sensitivity to LTS/EIS changes with time is not clear. The LCC sensitivity to LTS/EIS is assumed to be time-scale invariant to estimate the LTS/EIS-induced low cloud feedback (Klein et al., 2017).

This sentence has been changed as:
**Line 57-60 in the revised manuscript,** "The LCC on daily time scales is not as well explained by the LTS/EIS as the LCC on longer time scales. But the LCC sensitivity to LTS/EIS is assumed to be time-scale invariant to estimate the LTS/EIS-induced low cloud feedback and thus leads to some uncertainty (Klein et al., 2017). The explanation for the variant relationship between LCC and LTS/EIS across different time scales is not clear. And it is also not known whether the LTS and EIS can approximate the IS with the same accuracy across different time scales.".

*3.   Line 56-57: this sentence is unclear, especially "may not be conserved but with a stable layer". Please rephrase.*

**Response:** This sentence has been rephrased as:
**Line 62-65 in the revised manuscript**, "In deep decoupled PBLs, a strong stratification with a large $\theta$ increase between cloud layers and surface-mixed layers would exist (Jones et al., 2011; Nicholls, 1984). In this case, both the LTS and EIS likely count the stable layer of the decoupling into the IS estimates and thus overestimate the real IS atop the PBL.".

*4.   Line 70: "ARM" has not been introduced yet, what is it and which agency runs this? For the title of the subsection I suggest replacing "ARM sites" with "ground-based sites" or a short introduction of section 2 detailing the sort of data to be presented would be great*

*and it could be used to introduce ARM etc..*

**Response:** Atmospheric Radiation Measurement (ARM) was established by U.S Department of Energy Office of Biological and Environmental Research to provide an observational basis for studying the Earth's climate.

The title of the section 2.1 has been changed as:
**Line 81 in the revised manuscript,** "Radiosonde and cloud observations at the ground-based sites".

A brief overview of the sort of data to be presented has been added at the beginning of section 2:
**Line 76-80 in the revised manuscript, "**Data used in this study includes: (1) high vertical-resolution radiosondes and cloud radar and lidar observations from the ground site of Atmospheric Radiation Measurement (ARM) Program; (2) radiosondes of several subtropical and midlatitude stations from the Integrated Global Radiosonde Archive (IGRA) of the National Oceanic and Atmospheric Administration (NOAA); (3) global satellite observations of LCC; (4) the fifth-generation atmospheric reanalysis from the European Centre for Medium-Range Weather Forecasts (ECMWF). Methodologies of data processing are also introduced.**"**

5. *Line 71: so here "SGP" is used, I do not think it has ever been said it means Southern Great plains. Also why use this site? It is inland and therefore quite different from the environments usually associated with low clouds such as the subtropical stratocumulus regions. Why isn't ENA your primary site instead? This needs to be explained at the outset.*

**Response:** The definition of "SGP" has been added. The ENA site is located at the midlatitude ocean and here low stratus and stratocumulus clouds occur usually with no inversion (Norris, 1998) so that it is not an ideal site to investigate the relationship between LCC and IS. At the SGP, LCC and IS are closely related to each other but this LCC-IS relationship is not reflected by the LTS and EIS, which outstands the problem of using the LTS and EIS to estimate the IS.

The explanation about why the ENA isn't the primary site has been added at the beginning of the section 2.1:
**Line 86-90 in the revised manuscript,** "However, the ENA is located on Graciosa Island at the midlatitude ocean where low clouds frequently occur but with no inversion (Norris, 1998) so that it is not an ideal site to investigate the relationship of LCC with IS. Thus, the observations at the ENA are only used to validate the accuracy of EISp by comparing with the radiosonde-measured IS.".

6. *Line 72: "the ARM" is odd, just "ARM" would suffice. But it should really be "established by the Atmospheric Radiation Measurement program of the Department of Energy", unless it is already said in an intro (see comments above).*

**Response:** It has been clarified as:
**Line 83-84 in the revised manuscript,** "ARM was established by U.S Department of Energy

Office of Biological and Environmental Research to provide an observational basis for studying the Earth's climate.".

7. *Line 95: do you keep profiles that have low clouds but with other clouds aloft also present?*

**Response:** Yes, these profiles are kept and a sentence has been added as:

**Line 115 in the revised manuscript,** "Segments that have low clouds but with other clouds aloft are kept.".

8. *Line 97-99: here you mention separating coupled and decoupled PBL conditions but do not explain how it is done until much later in the paper. So either explain the method here or refrain from introducing the classification until it becomes relevant.*

**Response:** The method separating coupled cloudy, decoupled cloudy and clear sky PBL conditions have been explained in section 2.1:

**Line 119-127 in the revised manuscript,** "A straightforward indicator to distinguish coupled and decoupled PBLs is the height difference between the cloud base and the LCL ($\Delta z_b$) (Jones et al., 2011). When the PBL is well mixed, $\Delta z_b$ is close to zero, but in the decoupled PBLs the cloud and subcloud layers would be separated by a stable layer and the LCL may diverge from the cloud base hundreds of meters with large $\Delta z_b$ (Nicholls, 1984; Jones et al., 2011). The threshold value of $\Delta z_b$ is empirical and for different instrument capability, vertical resolution and locations the threshold may be a little different. In reference to the linear least-square fit between $\Delta z_b$ and $\Delta \theta$ in Jones et al. (2011) that 150 meters of $\Delta z_b$ correspond to 0.5K of the $\theta$ difference in the subcloud layer, a similar linear relationship is found but the slope is a little different that 180 meters of $\Delta z_b$ corresponds to 0.5K of the $\theta$ difference at the SGP site. Thus at the SGP site, a threshold value of 180 meters for $\Delta z_b$ is used to distinguish coupled and decoupled PBLs.".

9. *Line 102-103: the sentence "our results…" is unclear and incorrect. How about replacing with "this will be verified and discussed later".*

**Response:** Thanks. This sentence has been changed as:

**Line 130 in the revised manuscript,** "This will be verified and discussed later.".

10. *Lines 103-107: finally the reason for choosing SGP over ENA is explained but the explanation is confusing. And sounds very ad hoc and not very scientific. I think that your point is that the work focuses on the strong relationship between cloud cover and inversion strength, but this relationship is not true everywhere and in particular it is not observed at the ENA or weak. But it is found at the SGP, making this location a good place to explore a refinement of the inversion strength estimate. I wonder why the ENA site is mentioned at all?*

**Response:** We mentioned the ENA site because observations of radiosondes at the ENA still support that EISp is a better estimate for the IS than the LTS and EIS although here the LCC seems to be not controlled by the IS. These results are summarized and listed in Table 2. At the ENA, the IS-EISp correlation is 0.76, higher than that for LTS/EIS (0.51/0.58). Unfortunately, low clouds usually occur here without inversion so that it is not an ideal site to investigate the LCC-IS relationship. But for examining EISp and exploring why EISp is a

better estimate for the IS than LTS and EIS, the ENA still can provide long-term high-quality observations and we listed key information about the ENA PBL thermal structure deviating from the well-mixed conditions in Table 2.

Since this sentence is confusing, we have reorganized this section according to the reviewer's comments. We have clarified why we choose the SGP at the beginning of the section 2.1 and meanwhile we moved the detail analyses to the main text. See our responses to Comment 5.

*11. Line 115: another acronym that is not defined: "OWS"*
**Response:** OWS is ocean weather station and it was defined in introduction:
**Line 55 in the revised manuscript,** "at the subtropical ocean weather station (OWS) N".

*12. Line 170: "as the height of three fourth of the greatest …." Needs to be rewritten, it is really hard to understand what is done.*
**Response:** It has been rewritten as:
**Line 193-194 in the revised manuscript,** "the inversion top/base is defined as the nearest level above/below the layer of the maximum d$\theta$/dz where d$\theta$/dz equals to three-fourths of the maximum d$\theta$/dz.".

*13. Line 172: "these two methods both works" is incorrect, consider replacing with "these two methods work" or "these methods both work", BUT either way, it is confusing: does this mean the two methods give the same result? Which method do you actually use? Why mention two if you only use the one?*
**Response:** Yes, the two methods give the same result statistically. We actually used the first method. The sentence about the alternative method has been deleted.

*14. Line 176-179: the technique used to get the inversion strength for coarse vertical resolution radiosoundings is confusing, please clarify.*
**Response:** The technique used to get the inversion strength is the same as the EISp method, which adapted from the algorithm of obtaining the IS from high-resolution radiosondes, but is suitable for coarse-resolution atmospheric profiles of either soundings or reanalysis. Thus the method of getting the inversion strength for coarse-resolution data has been moved to section 3.1:

It has been clarified as:
**Line 226-230 in the revised manuscript.** "The EISp is designed to capture the IS information from the thinnest layer encompassing the inversion in low-resolution (hundreds of meters) atmospheric profiles. For these coarse-resolution profiles (e.g., ERA5), it is difficult to accurately locate the exact place of the inversion because usually the thickness of the inversion is much smaller than the distance between two adjacent vertical levels. Thus only one or two adjacent layers that could encompass the inversion are located. The latter is for the consideration that an inversion layer may be across two adjacent layers of the ERA5.".

*15. Line 188: replace "in consistent with that" with "as"*

**Response:** It has been changed as:
**Line 209 in the revised manuscript,** "as consistent with that".

*16. Line 193: as for section 2, a short intro in section 3 would help situate where the analysis is going next and why.*
**Response:** A short introduction has been added in section 3:
**Line 214-218 in the revised manuscript,** "In this section, the new EISp algorithm is established based on ground-based observations at the SGP and validated at other radiosonde stations of subtropics and midaltitude. In section 3.1, the new EISp algorithm is described. In section 3.2, at the SGP site with long-term 10m-resolution radiosondes, two questions are discussed: (1) why and how is EISp a better estimate for the IS than LTS and EIS? (2) how well does EISp control LCC as compared to LTS and EIS when it is a better estimate for the IS? In section 3.3, the EISp is further validated at radiosonde stations of subtropics and midlatitude.".

*17. Line 224: why use "180 m" and not "150m" as in Jones et al? not clearly explained.*
**Response:** The threshold of 150m in Jones et al is empirical, which should depend on the vertical resolution of data, instrument capability and locations. Thus, at the SGP site, this empirical threshold may be a little different. In reference to the linear least-square fit between $\Delta z$ and $\Delta \theta$ in Jones et al that $\Delta z$=150m corresponding to $\Delta \theta$=0.5K, we found similar linear relationship but the slope is a little different that $\Delta z$=180m corresponding to $\Delta \theta$=0.5K. Thus we use the threshold of 180m to distinguish coupled and decoupled PBLs.

Explanations have been added as:
**Line 119-127 in the revised manuscript,** "A straightforward indicator to distinguish coupled and decoupled PBLs is the height difference ($\Delta z$) between the cloud base and the LCL (Jones et al., 2011). The threshold value of $\Delta z$ is empirical and for different instrument capability, vertical resolution and locations the threshold may be a little different. In reference to the linear least-square fit between $\Delta z$ and $\Delta \theta$ in Jones et al. (2011) that 150 meters of $\Delta z$ correspond to 0.5K of the $\theta$ difference in the subcloud layer, a similar linear relationship is found but the slope is a little different that 180 meters of $\Delta z$ corresponds to 0.5K of the $\theta$ difference at the SGP site. Thus at the SGP site, a threshold value of 180 meters for $\Delta z$ is used to distinguish coupled and decoupled PBLs.".

*18. Line 225: so is this method the same as that used in Jones et al 2011 or something else? If it is then just say so. If it is not the same, then justify it.*
**Response:** It is the same as that used in Jones et al 2011 to use the height difference ($\Delta z$) between the cloud base and the LCL to determine coupled and decoupled PBLs, but the threshold value is slightly different (see our response to comment 17).

*19. Line 228: "large positive skewness": not sure I understand why this is? This should be discussed.*
**Response:** The large positive skewness means the value of $\theta_{LCL} - \theta_0$ can be large when the PBL is recognized as the coupled condition by the height difference ($\Delta z$) between the cloud base and the LCL. The exact reason of this is also not clear to us but observed at the

SGP. It may be related to whether the low cloud is actually formed by the lifting of local near ground air parcel. If the cloud is originated from other places but advected to here, it may be not necessarily related to the local PBL structure. Thus it could happen that the value of $\theta_{LCL} - \theta_0$ is large when $\Delta z$ is small. Also it is noted that the LCL of current near surface parcel is close to the low cloud base is only a necessary condition of a PBL being coupled. If a near surface stable layer has developed and decoupled with cloud layer, the LCL computed with the near surface air can still be close to the cloud base in coincident although the chance might be relatively small compared with coupled PBL.

Discussions have been added as:
**Line 288-292 in the revised manuscript,** "The exact reason of the positive skewness is not clear. Because the height of LCL being close to the simultaneously observed cloud base height is only a necessary condition of a PBL being coupled. A decoupled surface layer and overlaying cloud layer coincidently have the height of LCL close to the cloud base is not a surprise. Either clouds advected from other places or a new surface stable layer has developed while clouds formed earlier are still left above might result in positive $\theta_{LCL} - \theta_0$.".

*20. Line 230-231: this sentence is not entirely clear nor correct. By construction, LTS and EIS will include the term (theta_LCL-theta_0) regardless of its value. What you're trying to say I think is that in situations where this term is not zero, such as clear or decoupled PBLs, this will cause LTS and EIS to deviate from the real inversion strength value. No? Please rephrase accordingly.*
**Response:** Yes, that is what we are trying to say. This sentence has been rephrased as:
**Line 293-294 in the revised manuscript,** "Thus the non-zero term of $\theta_{LCL} - \theta_0$ will cause LTS and EIS to largely deviate from the real value of IS in the decoupled cloudy and clear sky segments.".

*21. Line 232: "even in the coupled PBLs: do you mean "decoupled" instead? Or is "even" misplaced? Unclear.*
**Response:** The sentence has been changed as:
**Line 295 in the revised manuscript,** "a premise of using LTS and EIS to measure the IS is that the lower-tropospheric $\theta$ gradient can be predicted by the moist adiabat above the LCL".

*22. Line 241: "could easily overwhelm the real IS" is weird. What you simply mean is that this term will cause an overestimate of the IS estimated with LTS or EIS, so why not rephrase this sentence with this in mind.*
**Response:** This sentence has been rephrased as:
**Line 302-304 in the revised manuscript,** "the term $\theta_{LCL} - \theta_0$ in Eqs. (9a) and (9b) will cause a strong overestimate of the IS by the LTS and EIS. And the variation of the LTS and EIS is attributed to not just variations of IS but also variations of the systematical deviations of temperature profiles from the dry adiabat below the LCL.".

*23. Line 259-260: this sentence seems very obvious, no clouds present suggests adverse conditions for cloud formation indeed. Not sure what the point is?*

**Response:** This sentence has been deleted.

*24. Figure 2 and discussion: this is for all seasons for the full period with SGP data, correct? Might want to remind the reader.*
**Response:** Yes, it is for all seasons for the full period with SGP data. This has been added to the caption of Figure 2:
**Line 317 in the revised manuscript,** "All composites are based on daily data of all seasons for the full period at the SGP.".

*25. Line 282-283: I wonder if the better relationship between RH and LTS compared to EIS does not come from the fact that EIS is meant to approximate inversion strength so necessitates that an inversion be present, when LTS is meant to represent static stability and can still relate to clouds when there is no inversion (or very weak). Have you tried to check LCC vs EIS by separating situations with IS >0 from those with IS=0 or <0?*
**Response:** It may be right about the LTS and EIS correlations with LCC are not fully caused by the inversion. The relationship of LCC with LTS and EIS is examined under different situation of whether inversion exists as suggested. Under situation of no inversion (IS<=0), the correlation of LCC with LTS and EIS is 0.05 (not significant) and -0.26. For situation of inversion (IS>0), the correlation of LCC with LTS and EIS is 0.34 and -0.01. For both situations LTS works better than EIS, but there is no evidence to approve that the LTS can still relate to clouds when inversion does not exist at SGP. And this also cannot explain the better relationship between LTS and LCC as compared to EIS. It is interesting that there is a significant negative correlation between EIS and LCC when there is no inversion. As discussed in the manuscript, the EIS can be written as $(\theta_{LCL} - \theta_0) + (\Delta\theta - \Gamma_m \Delta z) + \text{IS}$. When there is no inversion and if the free troposphere is moist adiabatic, EIS is actually the stratification below the LCL $\theta_{LCL} - \theta_0$ but not the inversion. As shown in Fig. 3b of the manuscript, the correlation between $\theta_{LCL} - \theta_0$ and LCC is -0.47 and this can explain the negative correlation of EIS when the inversion does not exist.

*26. Line 339: for the ERA5 vs SGP sonde comparison, what is the time resolution of the ERA5? 3-hourly or hourly? Is the same time of day used?*
**Response:** The time resolution of the ERA5 is hourly and the same time of day is used. This sentence has been changed as:
**Line 384 in the revised manuscript,** "The LTS, EIS and EISp derived from the hourly ERA5 reanalysis are directly compared against the SGP radiosonde-measured IS.".

*27. Line 343: "while decreasing RH…" is unclear, please clarify.*
**Response:** It has been rephrased as:
**Line 389 in the revised manuscript,** "with the EISp weakening and the inversion layer lifting, RH decreases but distributes to higher levels.".

*28. Line 346: language not very scientific, replace "does a better job on estimating the IS" with "offers a better fit to the real IS"*
**Response:** Thanks. It has been changed as:
**Line 392 in the revised manuscript,** "the EISp offers a better fit to the real IS".

*29. Line 373: language not very rigorous, replace "not bad" with e.g. "remarkably close to that explained by IS"*

**Response:** It has been changed as:

**Line 419 in the revised manuscript,** "remarkably close to that explained by IS".

*30. Section 4: again an intro would help follow the narrative.*

**Response:** An introduction has been added in the section 4:

**Line 474-481 in the revised manuscript,** "In this section, the relationship of global LCC with LTS, EIS and EISp is discussed through daily to seasonal time scales. Since ground-based observations of radiosondes from ARM and IGRA are all assimilated in the ERA5 reanalysis (Hersbach et al., 2020), it is not surprising that the assimilated output can well capture the PBL thermal structures to estimate the IS for these locations where ground-based observations are available. However, for most areas of oceans, only limited radiosondes are available over scattered islands or during short-term campaign of field experiments to be used in ERA5 assimilation and thus whether the IS can be right captured by the ERA5 profiles needs further examination. In this section, whether the EISp derived from the ERA5 profiles at the global scale (especially for oceans with few radiosondes assimilated into the ERA5) can better constrain LCC than LTS and EIS is explored.".

*31. Section 4.1, title not exact, only a few locations are chosen, this is not true worldwide, please rephrase.*

**Response:** The title has been changed as:

**Line 424 in the revised manuscript,** "at radiosonde stations of subtropics and midlatitude".

*32. Table 2, caption: what does "excluding 7-day means" mean?*

**Response:** It means 7-day means are subtracted from the data to isolate the daily IS-LCC relationships. This sentence has been rephrased as:

**Line 432 in the revised manuscript,** "The daily IS-LCC correlation is based on the data after subtracting 7-day means.".

*33. Line 390: "globally" is again too vast, maybe "in other locations" or "in maritime locations" would be more appropriate?*

**Response:** It has been changed as:

**Line 436 in the revised manuscript,** "at other radiosonde stations of subtropics and midaltitude".

*34. Line 391: here the coupled vs decoupled distinction is done again, and cloud base height is mentioned (as having been used earlier). This was not clear. Maybe best to always use the same definition no? could this alpha parameter be tested for the high resolution soundings as well?*

**Response:** We separate clear segments from cloudy segments because the occurrence frequency of clear sky segments best illustrates the failure of LTS/EIS on capturing PBL moisture and constraining LCC (Fig.3). However, when validating EISp at other stations the delta-z method is not accessible because the cloud base information is not available for

ground-based stations of IGRA, the method used at the SGP to distinguish the coupled-cloudy, decoupled-cloudy and clear sky segments is not accessible. Thus the alpha parameter is used to distinguish coupled and decoupled PBL according to the PBL thermal structures. This alpha parameter has been tested for the high-resolution soundings and it also works and come to similar results. In fact, results listed in Table 2 based on alpha parameter show consistent results with that based on delta-z.

Explanations have been added as:
**Line 438-440 in the revised manuscript,** "Because the cloud base height information is not available at the radiosonde stations of IGRA, the method used at the SGP to distinguish the coupled-cloudy, decoupled-cloudy and clear sky segments is not accessible. Thus, an alternative indicator, the decoupling degree ($\alpha_\theta$), is used to distinguish coupled and decoupled PBL according to the PBL thermal structures.";

**Line 454-456 in the revised manuscript,** "$\alpha_\theta$ has been tested for the high-resolution soundings and it also works and come to similar results. In fact, results listed in Table 2 at the SGP based on $\alpha_\theta$ show consistent results with that based on $\Delta z_b$.".

*35. Line 396: unclear, is this parameter introduced by Wood and Bretherton? Or by you but happens to be consistent to what they had done?*
**Response:** The definition of $\alpha_\theta$ is introduced in Wood and Bretherton (2004) but using the liquid potential temperature ($\theta_L$) as the conserved variable during the moist adiabat. Here, $\theta$ is used to construct the moist-adiabatic conserved variable by removing the moist-adiabatic $\theta$ increase above the LCL to express $\alpha_\theta$. A difference is $\alpha_\theta$ expressed by $\theta_L$ in Wood and Bretherton (2004) but expressed by $\theta$ here. The physical meaning is the same. Since IS and EIS are both expressed by $\theta$, by using $\theta$ to express $\alpha_\theta$, the EIS, IS and decoupling degree ($\alpha_\theta$) can be connected by a formula EIS=IS/$(1-\alpha_\theta)$, which is deduced in the revised manuscript (see our following response to the next comment about adding the explanation about the formula #12). This formula can further help to understand the relationship between EIS, IS and decoupling and what the EIS truly represents. This sentence has been rewritten and see our response to the next comment.

*36. Line 397: "the decoupling degree is in proportion…" is awkward. You simply mean that when alpha is zero the PBL is coupled, and when it is not, it is decoupled. Because a very small value suggest a state very close to coupled, you use 0.2 as a threshold to separate the two conditions. Please rewrite this paragraph in a similar fashion if my interpretation is correct. That being said, a physical explanation of what the formula (#12) is expressing is missing and should be added to help understand what it represents.*
**Response:** This paragraph has been rewritten and a physical explanation about formula (#12) has been added as:
**Line 441-454 in the revised manuscript,** "The definition of $\alpha_\theta$ is introduced in Wood and Bretherton (2004) by using the liquid potential temperature ($\theta_L$) as the conserved variable during the moist adiabat. Here, $\theta$ is used to construct the moist-adiabatic conserved variable by removing the moist-adiabatic $\theta$ increase above the LCL to express the $\alpha_\theta$

parameter:

$$\alpha_\theta = \frac{\theta_{ISB}-\theta_0-\Gamma_m(z_{ISB}-z_{LCL})}{\theta_{IST}-\theta_0-\Gamma_m(z_{IST}-z_{LCL})}. \text{ (12)}$$

The subscripts "ISB", "IST", "0", "700hPa" and "LCL" indicate the base and top of inversion layers, the levels of 1000hPa and 700hPa and the LCL, respectively. To understand its meaning, Eq. (12) can be transformed as:

$$\alpha_\theta = \frac{\theta_{LCL}-\theta_0+[\theta_{ISB}-\theta_{LCL}-\Gamma_m(z_{ISB}-z_{LCL})]}{IS+\theta_{LCL}-\theta_0+[\theta_{ISB}-\theta_{LCL}-\Gamma_m(z_{ISB}-z_{LCL})]} \approx \frac{EIS-IS}{EIS}. \text{ (13)}$$

The numerator of $\alpha_\theta$ can be understood as the strength of the PBL thermal structures deviating from the coupled conditions. The denominator of $\alpha_\theta$ can be understood as the sum of the deviation strength of the PBL thermal structure from the coupled conditions and the IS (or EIS). By Eq. (13), the EIS can also be expressed as $IS/(1-\alpha_\theta)$. Thus, whether the EIS is the real IS is actually determined by the decoupling parameter $\alpha_\theta$. In perfectly coupled conditions, $\alpha_\theta$ is zero and the EIS is exactly the IS. In decoupled PBLs, when $\alpha_\theta$ is larger, the EIS actually more accounts for the deviation of the PBL thermal structure from the coupled condition. A small value of $\alpha_\theta$ would suggest a state very close to the coupled condition and thus here a threshold value of 0.2 is used to distinguish the coupled/decoupled PBLs based on Eq. (12).".

*37. Line 424: I am not sure why the second sentence only mentions EISp, clearly there are no cloud data in high elevations regions, namely Tibetan Plateau and Andes (maybe others but these two are the most obvious) so just say so, the reader can see the analysis cannot be done in these areas for sure.*
**Response:** It has been changed as:
**Line 489 in the revised manuscript,** "For the LTS, EIS and EIS$_p$, the plateau regions with the surface pressure smaller than 700hPa are not investigated here, where no GEO-MODIS LCC is observed.".

*38. Line 425-432: this whole discussion of Fig 8 is unclear and hand-wavy at best. I cannot clearly see what the authors are trying to convey here. It might help to overlap the LCC contours (as solid lines) on the LTS/EIS/EISp maps and possibly draw boxes around the areas under focus here. But the whole paragraph needs clarification for sure.*
**Response:** This paragraph has been rewritten as:
**Line 490-506 in the revised manuscript,** "Overall, the annual mean value of LTS and EIS are obviously larger than the EISp value except the inner tropical convective zone where the EIS value is large negative. In addition, there are three differences between the spatial distributions of LTS, EIS and EISp:
(1) *Over the subtropical eastern oceans*, the center locations of LTS, EIS and EISp are different. For LTS and EIS, their center locations are more eastward and adjacent to the coast as compared with the center locations of EISp and LCC. For EISp, its center locations are relatively away from the coast and more consistent with the center locations of LCC.
(2) *Over midaltitude oceans*, the contrast of the values between the midlatitude and the tropics is different for LTS, EIS and EISp. The midaltitude LTS reduces to the minimum but still corresponds to about 40% of LCC. The midlatitude EIS is as strong as the EIS over the subtropical eastern oceans but corresponds to the LCC much less than the subtropical LCC. Only the variation of EISp from tropics to midlatitude is more reasonably consistent with the

spatial variation of LCC.

(3) *Over land*, the LTS and EISp explains over half of the LCC spatial variance according to their linear fit, but the EIS only explains 2% of the LCC spatial variance. This implies the IS is still a controlling factor for LCC distribution over land if using the LTS and EISp. The EIS barely correlates to continental LCC possibly because the EIS is strongly influenced by the term $\theta_{LCL} - \theta_0$ as discussed in section 3.

On the whole, the performance of EISp is better and less dependent on surface types. Over all global oceans and land, the EISp explains 78% of the spatial variance in LCC, significantly higher than that explained by the LTS (48%) and the EIS (13%). The spatial variations of LCC are also more sensitive to the EISp (Fig. 8f).".

The 60% LCC contour has been added on the LTS/EIS/EISp maps in Fig. 8.

*39. Line 436: "poor fault tolerance" is unclear, what does this mean?*
**Response:** this sentence has been rewritten as:
**Line 502 in the revised manuscript,** "The EIS barely correlates to continental LCC possibly because the EIS is strongly influenced by the term $\theta_{LCL} - \theta_0$ as discussed in section 3.".

*40. Line 438: language not very rigorous, consider replacing "but these will not happen with the EISp" with "The performance of EISp is less dependent on surface type" (assuming that this is what is meant).*
**Response:** The text has been rewritten as suggested:
**Line 504 in the revised manuscript,** "the performance of EISp is better and less dependent on surface types".

*41. Line 438-439: the whole sentence could equally say that LTS can explain LCC variations over land and EIS over ocean nearly as well as EISp. While it is clear EISp is performing better and more uniformly, your analysis still demonstrates where LTS and EIS have advantages and explains why they have been relied upon for quite some time.*
**Response:** This sentence has been deleted.

*42. Line 449/Fig 9: what is "all time scales" exactly??*
**Response:** All time scales mean the full daily time series. Daily, 7-day and monthly time scales mean the daily and 7-day window averaged anomalies and monthly means. This sentence has been changed as:
**Line 513-515 in the revised manuscript,** "the dependence of LCC on the LTS, EIS and EISp is further examined globally for the full daily time series (i.e., all time scales) and for the daily, 7-day window-averaged anomalies and monthly means (i.e., daily, 7-day and monthly time scales).".

*43. Line 459: you could add reference to the Klein et al. 2017 review that summarizes all the work done on the environmental factors that impact LCC in addition to IS. And this diversity explains the non geographical uniformity of the effect these parameters have on LCC, and in particular explains your result that "IS-LCC relationship is not uniform but varies with regions even over oceans". I would add that SST plays an important role and*

*it is not uniform either.*

**Response:** This reference has been added and the role of SST is also discussed as:

**Line 532-534 in the revised manuscript,** "Klein et al. (2017) also indicated that the LCC relationship with cloud controlling factors (e.g., the IS and sea surface temperature) is systematically different between the subtropical stratocumulus region and other regions (e.g., trade cumulus and midlatitude regions).".

*44. Line 468: I would suggest revisiting the work of Klein and Hartmann (1993) to better understand why LTS and LCC are related despite LTS not being well correlated with IS (again LTS is not a direct measure of inversion but of static stability).*

**Response:** This sentence has been changed as:

**Line 543-545 in the revised manuscript,** "the higher correlation of LTS with LCC might not be only attributed to the IS (LTS is not a direct measure of inversion but static stability). But more comprehensive and in-depth investigations on the LTS-LCC dependence are needed to understand the exact reason of this phenomenon.".

*45. Line 490: what is sensitivity here, the slope?*

**Response:** The sensitivity is the difference between the composited LCC values associated with the largest and smallest 10% of the LTS, EIS or EISp. This is defined and explained in Eq. 11. It has been changed as:

**Line 567 in the revised manuscript,** "the LCC sensitivity as defined in Eq. (11)".

*46. Lines 496-498: I do not understand the point of this last sentence, just that EISp is the only metric that exceeds 80% of the variance explained for all locations? Please consider simplifying the sentence.*

**Response: Yes,** this sentence has been changed as:

**Line 574 in the revised manuscript,** "Only the EISp can robustly explain the seasonal variance of LCC exceeding 80% for all locations.".

*47. Line 525: again the sentence "Unfortunately.." is unclear, what is this discussing exactly? is little known about this because it has not been explored or the authors do not have a satisfying explanation? "Limited knowledge" implies there is some, so what is it?*

**Response:** This sentence has been changed as:

**Line 602 in the revised manuscript,** "it has not been well explored about the midlatitude LCC-IS relationship.".

*48. Lines 529-530: it might be useful to remind the reader what the actual definition of LTS is and remind what KH93 had in mind. Also see Kawai and Teixeira (2010, JCLI, doi: 10.1175/2009JCLI3070.1).*

**Response:** A sentence has been added:

**Line 607-608 in the revised manuscript,** "the LTS not just includes the IS but actually represents the total static stability from 1000hPa to 700hPa to influence the amount and liquid water path of low clouds (Klein and Hartmann, 1993; Kawai and Teixeira, 2010)".

*49. Line 530-531: again, see comment above, IS is not the only cloud controlling factor, see*

*also Myers and Norris (2013, JCLI, doi:10.1175/JCLI-D-12-00736.1), cited in your manuscript.*

**Response:** This citation has been added:

**Line 611-613 in the revised manuscript,** "The IS is not the only LCC controlling factor, and other factors (e.g., SST, cold advection, free-tropospheric humidity and vertical velocity) are also important for influencing LCC (Myers and Norris, 2013; Klein et al., 2017).".

*50. Line 564: "questionable" should be replaced with "not recommended", but as conveyed in the comments above, there is a large amount of literature on the role of various meteorological metrics on cloud properties, I doubt that cloud feedback is estimated with only inversion strength in mind. But if it is, a citation should be added.*

**Response:** This sentence has been changed as:

**Line 643-645 in the revised manuscript,** "This non-uniform LCC sensitivity to cloud controlling factors across different regions and time scales suggests that using a single observational multi-linear regression between LCC and cloud-controlling factors to estimate the global low cloud feedbacks is not recommended.".

*Typos/grammar:*

*1. Line 101: remove "oceans", redundant.*
**Response:** It has been removed.

*2. Line 144: "away about 2.8km" is incorrect, replace with "within about 2.8 km"*
**Response:** It has been replaced.

*3. Line 148: here and in other places, "metrices" should be "metrics"*
**Response:** It has been corrected.

*4. Line 173: remove "s" at end of "jump" and "increase"*
**Response:** It has been removed.

*5. Line 208: "could compute" is odd, do you mean "would include"?*
**Response:** Yes, it has been replaced.

*6. Line 209: "as the IS estimates" is awkward, do you mean "in the IS estimates"?*
**Response:** Yes, it has been replaced.

*7. Line 209: Remove "in general", awkward.*
**Response:** It has been removed.

*8. Line 229: "strength" is not great, how about "value"?*
**Response:** Thanks. It has been replaced.

*9. Line 297: remove "that" at the beginning of this line.*
**Response:** It has been removed.

*10.Line 320: replace "three" after "among" with "them".*
**Response:** It has been replaced.

*11.Line 341: "close" should be "closer"*
**Response:** It has been corrected.

*12.Line 392: insert "parameter" at the end of the sentence (before the ":")*
**Response:** It has been inserted.

*13.Line 395&398: I believe "2004" should be "2006", unless there is a missing reference in the list.*
**Response:** It is Wood and Bretherton (2004), Boundary Layer Depth, Entrainment, and Decoupling in the Cloud-Capped Subtropical and Tropical Marine Boundary Layer, Journal of Climate, DOI: 10.1175/1520-0442(2004)017<3576:Bldead>2.0.Co;2.

*14.Line 488: "composites LCC to them" is odd, maybe "the slopes of the linear regression with LCC"?*
**Response:** It has been replaced.

*15. Line 489: "besides" should be "in addition" I think*
**Response:** It has been replaced.

*16. Line 499: "cross" should be "across"*
**Response:** It has been corrected.

*17. Line 529: "the worst estimation of the IS" should be "correlates the least with IS"*
**Response:** It has been corrected.

*18. Line 545: "wrong" should be "erroneous"*
**Response:** It has been corrected.

---

## Author Comment (AC2)

**Response to Anonymous Referee #2**

*Reviewer #2: I have reviewed the manuscript titled "A profile-based estimated inversion strength" submitted by Wang et al. to the Atmospheric Chemistry and Physics (ACP). The authors analyzed the radiosondes data at the Southern Great Plains (SGP) station and found the Inversion Strength estimation deviation in the current two variables, the Lower- Troposphere Stability (LTS) and Estimated Inversion Strength (EIS). Two sources of errors are identified, a systematic deviation from the dry adiabat below the lifting condensation level (LCL) and error from the spread of the actual potential temperature around the moist adiabat above the LCL. Based on the findings, the authors proposed a new measure/variable, profiled based inversion strength estimate (EISp) to estimate the inversion strength from the profiles in the reanalysis data. This new measure is defined as the potential temperature difference between the top and bottom of a layer with the maximum potential temperature gradient and with a subtraction of potential temperature changes due to the moist adiabat in this layer. Further analysis shows the EISp explains the low-level cloud significantly better in term of both spatial and temporal variations than the existing variables. The research presented in this manuscript is within the scope of ACP, and appears to significantly improve the understanding of low-level cloud changes by the proposed new measure. The new measure of the inversion strength is likely to be well accepted by the research community and extensively used in future studies. The manuscript is well written and easy to follow. Thus, I recommend the manuscript be accepted with minor revisions.*

**Response:** We thank the anonymous referee for reviewing our manuscript and very helpful comments to modify the manuscript. We have responded to all comments and carefully modified the manuscript accordingly.

*My major comments are:*

- *It appears the removal of the moist adiabat above the LCL does not benefit a better estimation of inversion strength in EIS based on the radiosonde analysis. Of course, this might be attributed to the spread of the inversion layer. It would be interesting to see if the EISp would actually perform better without the subtraction of the removal of the moist adiabat contribution, i.e. EISp as the potential temperature difference between the top and bottom of a layer with the maximum potential temperature gradient*

**Response:** Yes, it is interesting to try. By Fig. 2c and 2d (original Fig. 1c and 1d) in the revised manuscript, it is shown that the $\Delta\theta - \Gamma_m\Delta z$ distributions center at zero, which suggests that without removing the moist adiabat the distribution of $\Delta\theta$ likely has a positive shift because $\Gamma_m$ and $\Delta z$ are both positive. In addition, a wider spread is expected because of the variability in $\Gamma_m$ and $\Delta z$.

As suggested, the $\theta$ difference between the top and bottom of a layer with the maximum gradient ($\Delta\theta_{max}$) without the removal of the moist adiabat has been tested for its performance on constraining LCC based on the SGP observations. The correlation of $\Delta\theta_{max}$ with LCC is 0.11 and the LCC sensitivity to $\Delta\theta_{max}$ is 22%. In contrast, the correlation of EISp

with LCC is 0.34 and the LCC sensitivity to EISp is 50%. The correlation of $\Delta\theta_{max}$ with the radiosonde-measured IS is 0.47, which is also smaller than that of EISp (0.74). It seems to be necessary to remove the moist adiabat between the top and bottom of the layer with the maximum gradient to extract the IS from this layer.

- *It appears there are some limitations of the newly proposed variable. Firstly, it is for reanalysis data. For radiosondes with high vertical resolution, it seems the inversion strength can be directly calculated and might perform better than EISp. (Just curious, would the authors suggest the new variable for weather forecasting? It seems the LTS is commonly used in weather forecasting when only reanalysis data are available.). Secondly, all the test data sets are over extratropical regions and it is not clear how the controlling of low-level cloud compare over tropics with over extratropical regions, e.g. Figure 9. As the authors stated in the manuscript, this new variable may have limitations in the polar regions. The author could consider including the limitations this in the abstract.*

**Response:** For the first limitation, when the high-resolution radiosondes are available, the exact inversion top and base can be obtained fairly straightforward (Mohrmann et al., 2019). And in fact, the computation of EISp is adapted from the algorithm of obtaining the IS from high-resolution radiosondes, but is suitable for coarser resolution atmospheric profiles whether it is from the soundings or reanalysis. Because high-resolution soundings are rare, an applicable metric derived from reanalysis would be much more beneficial. It has been demonstrated in section 3 that the EISp can provide a more accurate estimate for the IS from the ERA5 profiles as compared to the LTS and EIS at the locations of subtropics and midlatitude with available radiosondes. Results in section 4 also demonstrate that the EISp is a better LCC controlling factor than LTS and EIS over subtropical eastern oceans. It has been explained in the main text:

**Line 251-254 in the revised manuscript**, "When high-resolution radiosondes are available, the exact IS can be obtained fairly straightforward (section 2.5, Eq. 6). The computation of EISp is in fact adapted from the algorithm of obtaining the IS from high-resolution radiosondes, but is adjusted to suit coarse-resolution atmospheric profiles in reanalysis. Because high-resolution soundings are rare, an applicable metric derived from reanalysis would be much more beneficial.".

For the second limitation, subtropics and midlatitude are most dominated by low clouds with the strongest cooling radiative effects on the earth (Klein and Hartmann, 1993) and thereby are the main focus for relationships between LCC and IS (e.g., Myers and Norris (2013), Qu et al. (2014), Mccoy et al. (2017) and Mohrmann et al. (2019)). Thus, these regions are the main focus of this work to provide a physically more reasonable and better cloud-controlling factor, which can be used to help evaluate the LCC responses to climate changes or aerosols. However, over tropics with less stratiform low clouds, the relationship between LCC and IS is not as important as those over subtropics and how the IS controls low clouds over tropics is also less important. Only a negative correlation between the LCC and IS is found and consistent with Szoeke et al. (2016) and the mechanism is still not clear. Since it is not an

important focus of investigating the relationship of LCC with IS, it is barely discussed in this work.

If the use of LTS in weather forecasting is intended to measure the IS, EISp might be a better choice since the EISp does capture the IS more accurately than the LTS especially on short time scales of 1-7 days according to our results. However, because LTS measures the bulk lower-tropospheric stability, whether EISp can be used to replace LTS for weather forecasting requires further investigation.

■ *The structure of the manuscript seems fine. The EISp definition can appear earlier in the manuscript. The authors could consider either moving section 3.1 to section 2, or making section 3.1 a separate section.*

**Response:** The EISp definition has been moved to section 3.1 to introduce the EISp at the beginning of the section 3:

**Line 226-255 in the revised manuscript,** "The EISp is designed to capture the IS information from the thinnest layer encompassing the inversion in low-resolution (hundreds of meters) atmospheric profiles. For these coarse-resolution profiles (e.g., ERA5), it is difficult to accurately locate the exact place of the inversion because usually the thickness of the inversion is much smaller than the distance between two adjacent vertical levels. Thus only one or two adjacent layers that could encompass the inversion are located. The latter is for the consideration that an inversion layer may be across two adjacent layers of the ERA5. Specifically, the EISp is computed as follows:

(1) Locating the layer of the maximum $\theta$ vertical gradient $(d\theta/dz)_{max}$:
For each hourly ERA5 profile, the layer of $(d\theta/dz)_{max}$ is firstly located between the LCL and 5km AGL (the red zone in Fig.1), since the inversion just features strong gradients in thermodynamical properties.

(2) Finding the layers encompassing the full inversion:
The layer of $(d\theta/dz)_{max}$ may not encompass the full inversion if the inversion crosses two adjacent layers of the ERA5. Thus, the layer of $(d\theta/dz)_{max}$ is combined with an adjacent layer just above or below it respectively, to constitute other two candidate layers that could encompass the full inversion (the blue and green zone in Fig.1).

(3) Calculating the EISp:
The EISp is calculated for the three possible layers identified in second stage, respectively:
$$EIS_p = \theta_{top} - \theta_{base} - \Gamma_m(z_{top} - z_{base}), (8)$$
where subscripts "top" and "base" represent the top and base levels of a candidate layer. $\Gamma_m$ is computed using Eq. (5) at the base level. The $\theta$ increase of the moist adiabat is removed to extract the strength of the inversion between the top and base levels, which is consistent with the EIS framework in Wood and Bretherton (2006). The final EISp is determined by which layer in Fig.1 encompasses stronger inversion computed from Eq. (8) and thus refers to the largest value among the three candidates EISp1-3.
The EIS (Wood and Bretherton, 2006) assumes that the PBL is well mixed (dry adiabat below the LCL and moist adiabat above the LCL) for estimating the IS. If that is the case, EISp would give the same results as EIS. However, it will be shown in the following sections that the actual PBL often deviates from the well mixed conditions, where the EISp provides

a physically more reasonable estimate for the IS than the EIS and thus a stronger cloud-controlling factor.

When high-resolution radiosondes are available, the exact IS can be obtained fairly straightforward (section 2.5, Eq. 6). The computation of EISp is in fact adapted from the algorithm of obtaining the IS from high-resolution radiosondes, but is adjusted to suit coarse-resolution atmospheric profiles in reanalysis. Because high-resolution soundings are rare, an applicable metric derived from reanalysis would be much more beneficial. Because the IGRA soundings have similar vertical resolutions as ERA5 in lower troposphere, the IS of these soundings (used in section 3.3) is derived exactly by the same way as the EISp.".

*Minor comments:*

- *Please spell out SGP (Southern Great Plains)*

**Response:** It has been corrected:

**Line 82 in the revised manuscript,** "Southern Great Plains".

- *L178, change "or" to "and"?*

**Response:** It has been changed:

**Line 237 in the revised manuscript,** "above and below it".

- *L215, this sentence only considers land case, please add the part for ocean case. L222, can you add explantion why the heigh difference can be used to differentiate the coupled and decoupled PBLs?*

**Response:** It has been added as:

**Line 276 in the revised manuscript,** "If over oceans, the levels of 3km and 150m can be replaced with 700hPa and 1000hPa.".

An explanation has been added as:

**Line 120-122 in the revised manuscript,** "When the PBL is well mixed, $\Delta z_b$ is close to zero, but in the decoupled PBLs the cloud and subcloud layers would be separated by a stable layer and the LCL may diverge from the cloud base hundreds of meters with large $\Delta z_b$ (Nicholls, 1984; Jones et al., 2011).".

- *L257, by eye check, it appears it is as sensitive as LTS.*

**Response:** This sentence has been changed as:

**Line 320 in the revised manuscript,** "the composites of LCC are slightly/significantly more sensitive to the changes of IS than the LTS/EIS.".

- *L261, I was just wondering whether the strong IS is the result of the cloud formation and then maintain the low level cloud?*

**Response:** Mauger and Norris (2010) found that stronger capping inversion at the PBL top precedes the increase in cloud cover. The stronger inversion reduces entrainment and decreases the rate at which dry air from above the inversion is incorporated into the

boundary layer, thus allowing the PBL to moisten and cloud cover to increase (Mauger and Norris, 2010). As soon as clouds form, the longwave radiation cools the cloud top and sharpens the inversion (Paluch and Lenschow, 1991).

■ *L288-L292, this part is really confusing and hard to understand.*
**Response:** It has been rewritten as**:**
**Line 346-359 in the revised manuscript,** "To understand why this happens, the LTS and EIS in Eq. (9) both have been separated into three terms to discuss. For the LTS, the two terms $\theta_{LCL} - \theta_0$ and $\Delta\theta$ of Eq. (9a) usually offset each other with a negative correlation of -0.56 and a slope of the least-squares fit -0.5K/K (Fig. 4a). In contrast, the slope of the least-squares fit between $\Delta\theta - \Gamma_m\Delta z$ and $\theta_{LCL} - \theta_0$ is only -0.05K/K (not shown). Furthermore, the LTS and EIS equation can be transformed into:
$$\text{LTS} = \left(1 + \frac{\Delta\theta}{\theta_{LCL} - \theta_0}\right)(\theta_{LCL} - \theta_0) + \text{IS, (10a)}$$
$$\text{EIS} = \left(1 + \frac{\Delta\theta - \Gamma_m\Delta z}{\theta_{LCL} - \theta_0}\right)(\theta_{LCL} - \theta_0) + \text{IS. (10b)}$$
On average, the coefficient before $\theta_{LCL} - \theta_0$ for the LTS in Eq. (10a) is 0.5 while that for EIS in Eq. (10b) is 0.95. The variation of LTS and EIS result from both the changes of IS (positively correlated with LCC as shown in Fig. 3c) and the changes of $\theta_{LCL} - \theta_0$ (negatively correlated with LCC as shown in Fig. 4b). According to Eqs. (10a) and (10b), the LTS actually only involves half of the bias caused by $\theta_{LCL} - \theta_0$ and thus not as strongly influenced by $\theta_{LCL} - \theta_0$ as the EIS. As a result, only removing the moist adiabat ($\Gamma_m\Delta z$) does not make the EIS a better estimate for the IS at the SGP but make the EIS more influenced by $\theta_{LCL} - \theta_0$. This explains why the LTS is better correlated with LCC and RH (Figs. 3a and 3d) than the EIS (Figs. 3b and 3e) at the SGP. However, the physical reason that why the PBL stratification changes in this way is unclear to us and it is beyond the scope of this study.".

■ *L293, this paragraph is confusing and hard to understand.*
**Response:** It has been rewritten as:
**Line 362-367 in the revised manuscript,** "Figs. 4c and 4d suggest that the spread of $|\Delta\theta - \Gamma_m\Delta z|$ increases with the layer thickness either between the LCL and the inversion base or between the inversion top and 3km AGL (with a correlation of 0.59 or 0.58, respectively). Thus, the thicker the layer encompassing inversion involved in the EIS calculation is, the larger the uncertainty is. Including more layers around the inversion layer in estimating the IS likely results in more uncertainty. This suggests a possible way of better estimating the IS if we can reduce the layer thickness ($\Delta z$) associated with the second term on the rhs of Eq. (9b), which also makes the IS estimate less dependent on the moist adiabatic assumption.".

■ *L312, for extreme case, e.g. radiosonde with very high vertical resolution, the EISp is the Inversion Strength? If this is the case, the authors might want to state this in the manuscript.*
**Response:** Yes, if using the high-resolution radiosondes with the exact inversion top and base, the EISp is just the IS. This has been stated in the manuscript:
**Line 252 in the revised manuscript,** "The computation of EISp is in fact adapted from the

algorithm of obtaining the IS from high-resolution radiosondes, but is suitable for coarser atmospheric profiles of either soundings or reanalysis.".

■ *Fig 9, why these is negative values for R-square? This figure shows correlation rather than R-square?*
**Response:** R-square is used with a minus/plus sign to indicate negative/positive correlations. It has been stated in section 2.6 and also mentioned in the caption of Fig.9:
**Line 511 in the revised manuscript,** "The minus/plus sign of R-square indicates negative/positive correlations.".

■ *Fig 9, The quality of text in this figure seems low compared to other figures.*
**Response:** The discussion about Fig.9 has been separated into three parts: (1) subtropical eastern oceans; (2) midlatitude oceans; (3) land regions.

The text has been rewritten as:
**Line 513-545 in the revised manuscript,** "In Fig.9, the dependence of LCC on the LTS, EIS and EISp is further examined globally for the full daily time series (i.e., all time scales) and for the daily, 7-day window-averaged anomalies and monthly means (i.e., daily, 7-day and monthly time scales). It is noted that the dependence of LCC on the three ERA5-based metrics are variant across different regions. LCC is best correlated with the three metrics over the subtropical eastern oceans and some land regions that are most dominated by low clouds. Over midlatitude oceans and inner tropical convergence zone, the LCC is weakly or negatively correlated with the three metrics. Thus, it is discussed separately for the most LCC-dominated regions over subtropical oceans, midlatitude oceans and land.
(1) Over the subtropical eastern oceans with more than 60% of LCC, on all time scales (Figs. 9a-c), the EISp explains 36% of the variance in LCC on average, larger than that explained by the LTS (21%) and the EIS (20%). The fact that EIS does not provide a stronger correlation with LCC than LTS was also recognized by Park and Shin (2019) and Cutler et al. (2022). In contrast, the explained variance of the linear fitting between LCC and EISp is 1.8 times of that with LTS and EIS. Besides, the mean LCC sensitivity (defined in Eq. (11) and not shown in the figure) to the EISp on all time scales is 48% over these regions, significantly higher than that to the LTS (37%) and the EIS (36%). Although radiosondes are rare and the ERA5 profiles are mostly from the model output over these regions, the EISp still provides a much stronger constraint on LCC than LTS and EIS. As shown in Figs. 9d-i through daily to monthly time scales, the EISp robustly explains larger LCC variance than the LTS and EIS especially on short time scales.
(2) Over midlatitude oceans, weak and not significant correlations between LCC and the three metrics exist through all of time scales in Fig.9. This poor relationship is also found at the ENA site (Table 2) even using the radiosonde to derive the IS, and thus it is not caused by using the ERA5 to estimate the IS. This suggests that the IS-LCC relationship is indeed not uniform but varies with regions. Klein et al. (2017) also indicated that the LCC relationship with cloud controlling factors (e.g., the IS and sea surface temperature) is systematically different between the subtropical stratocumulus region and other regions (e.g., trade cumulus and midlatitude regions). Thus, when the IS is used to constrain the environmental influence on LCC variations, it should be noted that LCC is not all uniformly constrained by the IS for

different regions. For some regions such as midlatitude oceans, the IS might not be a good constraint on LCC. But by more accurately estimating the IS, the EISp is more correlated with LCC than the LTS and EIS over midlatitude oceans such as North Pacific and North Atlantic on all time scales in Figs. 9a-c.

(3) Over land regions of relatively more LCC (about 15%-25% at south America, China and Europe), the correlation between EISp and LCC is comparable to the subtropical oceanic regions through all of time scales in Fig.9. This suggests the EISp is also an important controlling factor for continental LCC over these regions. Besides, the EISp is more correlated with LCC than the LTS and EIS over most land regions, except over China where the LTS explains larger LCC variance than the EIS and EISp. The higher correlation of LTS with LCC over China might not be only attributed to the IS (LTS is not a direct measure of inversion but static stability). But more comprehensive and in-depth investigations on the LTS-LCC dependence are needed to understand the exact reason of this phenomenon.**".

■ *L448, please explain how you calculate the values for different time scales, in section 2?*

**Response:** The method of separating different time scales has been explained in the section 2.6:

**Line 208-212 in the revised manuscript,** "for isolating the correlation and the regression slope on different time scales, window anomalies are defined as consistent with that in Szoeke et al. (2016): $x^{\Delta_i} = [x]^{\Delta_i} - [x]^{\Delta_{i+1}}$. The brackets represent mean of x over the window of length $\Delta$. The superscripts $\Delta_i$ and $\Delta_{i+1}$ are the i-th window length and the next longer window length.".

This sentence has been changed as:

**Line 513-514 in the revised manuscript,** "the dependence of LCC on the LTS, EIS and EISp is further examined globally for the full daily time series (i.e., all time scales) and for the daily, 7-day window-averaged anomalies and monthly means (i.e., daily, 7-day and monthly time scales).".

■ *L454, it appears the values are negative over tropical oceans.*

**Response:** This sentence has been changed as:

**Line 516-517 in the revised manuscript,** "LCC is best correlated with three metrics over the subtropical eastern oceans and some land regions that are most dominated by low clouds. Over midlatitude oceans and inner tropical convergence zone, the LCC is weakly or negatively correlated with the three metrics.".

■ *Figure 10, curious why use for example "Californian" than "California" or "Californian region"?*

**Response:** These have been changed as "Peru", "Namibia", "California", "Australia", "Canaria" in Figure 7 and Figure 10.

■ *L482, should be explained variance rather than "variance"? also L516.*

**Response:** The "variance" has been changed as "window-averaged LCC anomalies". The sentence has been changed as:

**Line 558-560 in the revised manuscript,** "As shown in Fig. 10 (the dash line in the left panel), over the five key tropical and subtropical eastern oceans, the daily and seasonal window-averaged LCC anomalies accounts for a larger portion of the total LCC variance, indicating the LCC variation mainly happens at the daily and seasonal time scales.".

The sentence at L516 has been changed as:

**Line 593 in the revised manuscript,** "67% of the LCC variance is from the daily time scale".